# Geometric Transformers for Protein Interface Contact Prediction

**Alex Morehead, Chen Chen, & Jianlin Cheng**
Department of Electrical Engineering & Computer Science
University of Missouri
Columbia, MO 65211, USA
{acmwhb,chen.chen,chengji}@missouri.edu

## Abstract

Computational methods for predicting the interface contacts between proteins come highly sought after for drug discovery as they can significantly advance the accuracy of alternative approaches, such as protein-protein docking, protein function analysis tools, and other computational methods for protein bioinformatics. In this work, we present the Geometric Transformer, a novel geometry-evolving graph transformer for rotation and translation-invariant protein interface contact prediction, packaged within DeepInteract, an end-to-end prediction pipeline. DeepInteract predicts *partner-specific* protein interface contacts (i.e., inter-protein residue-residue contacts) given the 3D tertiary structures of two proteins as input. In rigorous benchmarks, DeepInteract, on challenging protein complex targets from the 13th and 14th CASP-CAPRI experiments as well as Docking Benchmark 5, achieves 14% and 1.1% top L/5 precision (L: length of a protein unit in a complex), respectively. In doing so, DeepInteract, with the Geometric Transformer as its graph-based backbone, outperforms existing methods for interface contact prediction in addition to other graph-based neural network backbones compatible with DeepInteract, thereby validating the effectiveness of the Geometric Transformer for learning rich relational-geometric features for downstream tasks on 3D protein structures.[1]

## 1 Introduction

Interactions of proteins often reflect and directly influence their functions in molecular processes, so understanding the relationship between protein interaction and protein function is of utmost importance to biologists and other life scientists. Here, we study the residue-residue interaction between two protein structures that bind together to form a binary protein complex (i.e., dimer), to better understand how these coupled proteins will function *in vivo*. Predicting where two proteins will interface *in silico* has become an appealing method for measuring the interactions between proteins since a computational approach saves time, energy, and resources compared to traditional methods for experimentally measuring such interfaces (Wells & McClendon (2007)). A key motivation for determining these interface contacts is to decrease the time required to discover new drugs and to advance the study of newly designed proteins (Murakami et al. (2017)).

Existing approaches to interface contact prediction include classical machine learning and deep learning-based methods. These methods traditionally use hand-crafted features to predict which inter-chain pairs of amino acid residues will interact with one another upon the binding of the two protein chains, treating each of their residue pairs as being independent of one another. Recent work on interface prediction (Liu et al. (2020)), however, considers the biological insight that the interaction between two inter-chain residue pairs depends not only on the pairs' features themselves but also on other residue pairs ordinally nearby in terms of the protein complex's sequence. As such, the problem of interface contact prediction became framed as one akin to image segmentation

---

[1] Training and inference code as well as pre-trained models are available at https://github.com/BioinfoMachineLearning/DeepInteract

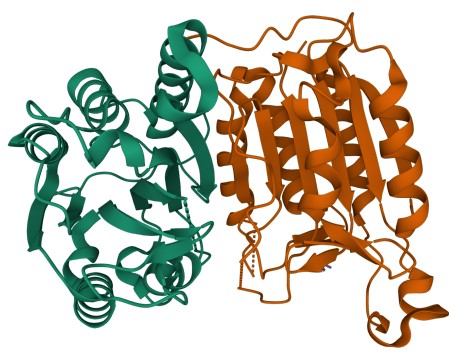

Figure 1: A Mol* (Sehnal et al. (2021)) visualization of interacting protein chains (PDB ID: 3H11).

or object detection, opening the door to innovations in interface contact prediction by incorporating the latest techniques from computer vision.

Nonetheless, up to now, no works on *partner-specific* protein interface contact prediction have leveraged two recent innovations to better capture geometric shapes of protein structures and long-range interactions between amino acids important for accurate prediction of protein-protein interface contacts: (1) geometric deep learning for *evolving* proteins' geometric representations and (2) graph-based self-attention similar to that of Vaswani et al. (2017). Towards this end, we introduce Deep-Interact, an end-to-end deep learning pipeline for protein interface prediction. DeepInteract houses the Geometric Transformer, a new graph transformer designed to exploit protein structure-specific geometric properties, as well as a dilated convolution-based interaction module adapted from Chen et al. (2021) to predict which inter-chain residue pairs comprise the interface between the two protein chains. In response to the exponential rate of progress being made in predicting protein structures *in silico*, we trained DeepInteract end-to-end using DIPS-Plus (Morehead et al. (2021)), to date the largest feature-rich dataset of protein complex structures for machine learning of protein interfaces, to close the gap on a proper solution to this fundamental problem in structural biology.

## 2 RELATED WORK

Over the past several years, geometric deep learning has become an effective means of automatically learning useful feature representations from structured data (Bronstein et al. (2021)). Previously, geometric learning algorithms like convolutional neural networks (CNNs) and graph neural networks (GNNs) have been used to model molecules and to predict protein interface contacts. Schütt et al. (2017) introduced a deep tensor neural network designed for molecular tasks in quantum chemistry. Fout et al. (2017) designed a siamese GNN architecture to learn weight-tied feature representations of residue pairs. This approach, in essence, processes subgraphs for each residue in each complex and aggregates node-level features locally using a nearest-neighbors approach. Since this *partner-specific* method derives its training dataset from Docking Benchmark 5 (DB5) (Vreven et al. (2015)), it is ultimately data-limited. Townshend et al. (2019) represent interacting protein complexes by voxelizing each residue into a 3D grid and encoding in each grid entry the presence and type of the residue's underlying atoms. This *partner-specific* encoding scheme captures static geometric features of interacting complexes, but it is not able to scale well due to its requiring a computationally-expensive spatial resolution of the residue voxels to achieve good results.

Continuing the trend of applying geometric learning to protein structures, Gainza et al. (2020) developed MaSIF to perform *partner-independent* interface region prediction. Likewise, Dai & Bailey-Kellogg (2021) do so with an attention-based GNN. These methods learn to perform binary classification of the residues in both complex structures to identify regions where residues from both complexes are likely to interact with one another. However, because these approaches predict *partner-independent* interface regions, they are less likely to be useful in helping solve related tasks such as drug-protein interaction prediction and protein-protein docking (Ahmad & Mizuguchi

(2011)). Liu et al. (2021a) created a graph neural network for predicting the effects of mutations on protein-protein binding affinities, and, more recently, Costa et al. (2021) introduced a Euclidean equivariant transformer for protein docking. Both of these methods may benefit from the availability of precise interface predictors by using them to generate contact maps as input features.

To date, one of the best result sets obtained by any model for protein interface contact prediction comes from Liu et al. (2020) where high-order (i.e. sequential and coevolution-based) interactions between residues are learned and preserved throughout the network in addition to static geometric features initially embedded in the protein complexes. However, this work, like many of those preceding it, undesirably maintains the trend of reporting model performance in terms of the median area under the receiver operating characteristic which is not robust to extreme class imbalances as often occur in interface contact prediction. In addition, this approach is data-limited as it uses the DB5 dataset and its predecessors to derive both its training data and makes use of only each residue's carbon-alpha ($C\alpha$) atom in deriving its geometric features, ignoring important geometric details provided by an all-atom view of protein structures.

Our work builds on top of prior works by making the following contributions:

- We provide the *first* example of graph self-attention applied to protein interface contact prediction, showcasing its effective use in learning representations of protein geometries to be exploited in downstream tasks.

- We propose the new *Geometric Transformer* which can be used for tasks on 3D protein structures and similar biomolecules. For the problem of interface contact prediction, we train the Geometric Transformer to evolve a geometric representation of protein structures simultaneously with protein sequence and coevolutionary features for the prediction of inter-chain residue-residue contacts. In doing so, we also demonstrate the merit of the recently-released Enhanced Database of Interacting Protein Structures (DIPS-Plus) for interface prediction (Morehead et al. (2021)).

- Our experiments on challenging protein complex targets demonstrate that our proposed method, *DeepInteract*, achieves state-of-the-art results for interface contact prediction.

## 3 DATASETS

The current opinion in the bioinformatics community is that protein sequence features still carry important higher-order information concerning residue-residue interactions (Liu et al. (2020)). In particular, the residue-residue coevolution and residue conservation information obtained through multiple sequence alignments (MSAs) has been shown to contain powerful information concerning intra-chain and even inter-chain residue-residue interactions as they yield a compact representation of residues' coevolutionary relationships (Jumper et al. (2021)).

Keeping this in mind, for our training and validation datasets, we chose to use DIPS-Plus (Morehead et al. (2021)), to our knowledge the largest feature-rich dataset of protein complexes for protein interface contact prediction to date. In total, DIPS-Plus contains 42,112 binary protein complexes with positive labels (i.e., 1) for each inter-chain residue pair that are found within 6 Å of each other in the complex's bound (i.e., structurally-conformed) state. The dataset contains a variety of rich residue-level features: (1) an 8-state one-hot encoding of the secondary structure in which the residue is found; (2) a scalar solvent accessibility; (3) a scalar residue depth; (4) a $1 \times 6$ vector detailing each residue's protrusion concerning its side chain; (5) a $1 \times 42$ vector describing the composition of amino acids towards and away from each residue's side chain; (6) each residue's coordinate number conveying how many residues to which the residue meets a significance threshold, (7) a $1 \times 27$ vector giving residues' emission and transition probabilities derived from HH-suite3 (Steinegger et al. (2019)) profile hidden Markov models constructed using MSAs; and (8) amide plane normal vectors for downstream calculation of the angle between each intra-chain residue pair's amide planes.

To compare the performance of DeepInteract with that of state-of-the-art methods, we select 32 homodimers and heterodimers from the test partition of DIPS-Plus to assess each method's competency in predicting interface contacts. We also evaluate each method on 14 homodimers and 5 heterodimers with PDB structures publicly available from the 13th and 14th sessions of CASP-CAPRI (Lensink et al. (2019), Lensink et al. (2021)) as these targets are considered by the bioinformatics

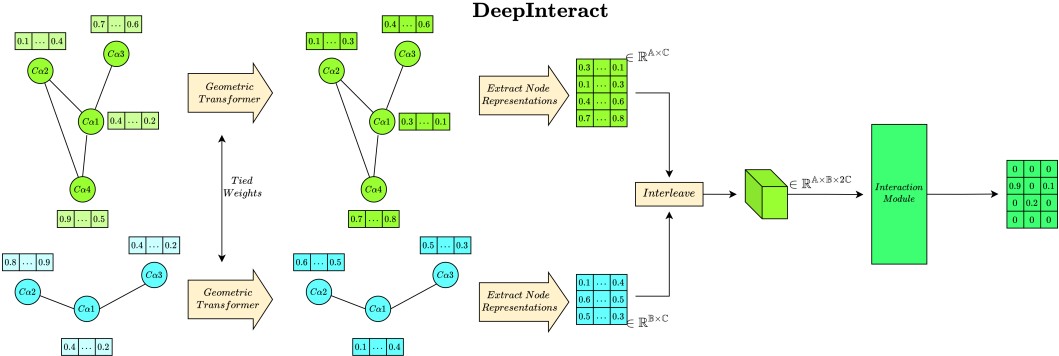

Figure 2: DeepInteract overview. Our proposed pipeline separates interface contact prediction into two tasks: (1) learning new node representations $h_{\mathbb{A}}$ and $h_{\mathbb{B}}$ for pairs of residue protein graphs and (2) convolving over $h_{\mathbb{A}}$ and $h_{\mathbb{B}}$ interleaved together to predict pairwise contact probabilities.

community to be challenging for existing interface predictors. For any CASP-CAPRI test complexes derived from multimers (i.e., protein complexes that can contain more than two chains), to represent the complex we chose the pair of chains with the largest number of interface contacts. Finally, we use the traditional 55 test complexes from the DB5 dataset (Fout et al. (2017); Townshend et al. (2019); Liu et al. (2020)) to benchmark each heteromer-compatible method.

To expedite training and validation and to constrain memory usage, beginning with all remaining complexes not chosen for testing, we filtered out all complexes where either chain contains fewer than 20 residues and where the number of possible interface contacts is more than $256^2$, leaving us with an intermediate total of 26,504 complexes for training and validation. In DIPS-Plus, binary protein complexes are grouped into shared directories according to whether they are derived from the same parent complex. As such, using a *per-directory* strategy, we randomly designate 80% of these complexes for training and 20% for validation to restrict overlap between our cross-validation datasets. After choosing these targets for testing, we then filter out complexes from our training and validation partitions of DIPS-Plus that contain any chain with over 30% sequence identity to any chain in any complex in our test datasets. This threshold of 30% sequence identity is commonly used in the bioinformatics literature (Jordan et al. (2012), Yang et al. (2013)) to prevent large evolutionary overlap between a dataset's cross-validation partitions. However, most existing works for interface contact prediction do not employ such filtering criteria, so the results reported in these works may be over-optimistic by nature. In performing such sequence-based filtering, we are left with 15,618 and 3,548 binary complexes for training and validation, respectively.

## 4 METHODS

### 4.1 PROBLEM FORMULATION

Summarized in Figure 2, we designed DeepInteract, our proposed pipeline for interface contact prediction, to frame the problem of predicting interface contacts *in silico* as a two-part task: The first part is to use attentive graph representation learning to inductively learn new node-level representations $h_{\mathbb{A}} \in \mathbb{R}^{\mathbb{A} \times \mathbb{C}}$ and $h_{\mathbb{B}} \in \mathbb{R}^{\mathbb{B} \times \mathbb{C}}$ for a pair of graphs representing two protein chains. The second part is to channel-wise interleave $h_{\mathbb{A}}$ and $h_{\mathbb{B}}$ into an interaction tensor $\mathbb{I} \in \mathbb{R}^{\mathbb{A} \times \mathbb{B} \times 2\mathbb{C}}$, where $\mathbb{A} \in \mathbb{R}$ and $\mathbb{B} \in \mathbb{R}$ are the numbers of amino acid residues in the first and second input protein chains, respectively, and $\mathbb{C} \in \mathbb{R}$ is the number of hidden channels in both $h_{\mathbb{A}}$ and $h_{\mathbb{B}}$. We use interaction tensors such as $\mathbb{I}$ as input to our interaction module, a convolution-based dense predictor of inter-graph node-node interactions. We denote each protein chain in an input complex as a graph $\mathbb{G}$ with edges $\mathbb{E}$ between the $k$-nearest neighbors of its nodes $\mathbb{N}$, with nodes corresponding to the chain's amino acid residues represented by their $C\alpha$ atoms. In this setting, we let $k = 20$ as we observed favorable cross entropy loss on our validation dataset with this level of connectivity. We note that this level of graph connectivity has also proven to be advantageous for prior works developing

**Geometric Transformer**

Figure 3: Geometric Transformer overview. Notably, our final layer of the Geometric Transformer removes the edge update path since, in our formulation of interface prediction, only graph pairs' node representations $h_{\mathbb{A}}$ and $h_{\mathbb{B}}$ are directly used for the final interface contact prediction.

deep learning approaches for graph-based protein representations (Fout et al. (2017); Ingraham et al. (2019)).

## 4.2 GEOMETRIC TRANSFORMER ARCHITECTURE

Hypothesizing that a self-attention mechanism that evolves proteins' physical geometries is a key component missing from existing interface contact predictors, we propose the Geometric Transformer, a graph neural network explicitly designed for capturing and iteratively *evolving* protein geometric features. As shown in Figure 3, the Geometric Transformer expands upon the existing Graph Transformer architecture (Dwivedi & Bresson (2021)) by introducing (**1**) an edge initialization module, (**2**) an edge-wise positional encoding (EPE), and (**3**) a geometry-evolving conformation module employing repeated geometric feature gating (GFG) (see Sections A.6, A.7, and A.8 for rationale). Moreover, the Geometric Transformer includes subtle architectural enhancements to the original Transformer architecture (Vaswani et al. (2017)) such as moving the network's first normalization layer to precede any affinity score computations for improved training stability (Hussain et al. (2021)). To our knowledge, the Geometric Transformer is the *first* deep learning model that applies multi-head attention to the task of *partner-specific* protein interface contact prediction. The following sections serve to distinguish our new Geometric Transformer from other Transformer-like architectures by describing its new neural network modules for geometric self-attention.

### 4.2.1 EDGE INITIALIZATION MODULE

To accelerate its training, the Geometric Transformer first embeds each edge $e \in \mathbb{E}$ with the initial edge representation

$$c_{ij} = \phi_e^1([\mathrm{p}_1 \,||\, \mathrm{p}_2 \,||\, \phi_e^{\mathrm{m}_{ij}}(\mathrm{m}_{ij} \,||\, \lambda_e) \,||\, \phi_e^{\mathrm{f}_1}(\mathrm{f}_1) \,||\, \phi_e^{\mathrm{f}_2}(\mathrm{f}_2) \,||\, \phi_e^{\mathrm{f}_3}(\mathrm{f}_3) \,||\, \phi_e^{\mathrm{f}_4}(\mathrm{f}_4)]) \tag{1}$$

$$e_{ij} = \phi_e^2(\rho_e^a(\rho_e^g(c_{ij}))) \tag{2}$$

where $\phi_e^i$ refers to the $i$'th edge information update function such as a multi-layer perceptron; $||$ denotes channel-wise concatenation; $\mathrm{p}_1$ and $\mathrm{p}_2$, respectively, are trainable one-hot vectors indexed by $P_i$ and $P_j$, the positions of nodes $i$ and nodes $j$ in the chain's underlying amino acid sequence; $\mathrm{m}_{ij}$ are any user-predefined features for $e$ (in our case the normalized Euclidean distances between nodes $i$ and nodes $j$); $\lambda_e$ are *edge-wise* sinusoidal positional encodings $sin(P_i - P_j)$ for $e$; $\mathrm{f}_1$, $\mathrm{f}_2$, $\mathrm{f}_3$,

**Geometric Transformer**
*Conformation Module*

Figure 4: Conformation module overview. The Geometric Transformer uses a conformation module in each layer to evolve proteins graphs' geometric representations via repeated gating and a final series of residual connection blocks.

and $f_4$, in order, are the four protein-specific geometric features defined in Section A.3; and $\rho_e^a$ and $\rho_e^g$ are feature addition and channel-wise gating functions, respectively.

### 4.2.2 CONFORMATION MODULE

The role of the Geometric Transformer's subsequent conformation module, as illustrated in Figure 4, is for it to learn how to iteratively *evolve* geometric representations of protein graphs by applying repeated gating to our initial edge geometric features $f_1$, $f_2$, $f_3$, and $f_4$. To do so, the conformation module updates $e_{ij}$ by introducing the notion of a *geometric neighborhood* of edge $e$, treating $e$ as a pseudo-node. Precisely, $\mathbb{E}_k$, the edge geometric neighborhood of $e$, is defined as the $2n$ edges

$$\mathbb{E}_k = \{e_{n_1 i}, e_{n_2 j} \mid (n_1, n_2 \in \mathbb{N}_k) \text{ and } (n_1, n_2 \neq i, j)\}, \tag{3}$$

where $\mathbb{N}_k \subset \mathbb{N}$ are the source nodes for incoming edges on edge $e's$ source and destination nodes. The intuition behind updating each edge according to its $2n$ nearest neighboring edges is that the geometric relationship between a residue pair, described by their mutual edge's features, can be influenced by the physical constraints imposed by proximal residue-residue geometries. As such, we use these nearby edges during geometric feature updates. In the conformation module, the iterative processing of all geometric neighborhood features for edge $e$ can be represented as

$$O_{ij} = \sum_{k \in \mathbb{E}_k} [(\phi_e^n(e_{ij,k}^n) \odot \phi_e^{f_n}(f_n)), \forall n \in \mathbb{F}] \tag{4}$$

$$e_{ij} = 2 \times ResBlock_2(\phi_e^5(e_{ij}) + 2 \times ResBlock_1(\phi_e^5(e_{ij}) + O_{ij})), \tag{5}$$

where $\mathbb{F}$ are the indices of the geometric features $\{f_1, f_2, f_3, f_4\}$ defined in Section A.3; $\odot$ is element-wise multiplication; $e_{ij,k}^n$ is neighboring edge $e_k$'s representation after gating with $f_{n-1}$; and $2 \times ResBlock_i$ represents the $i$'th application of two unique, successive residual blocks, each defined as $ResBlock(x) = \phi_e^{Res_2}(\phi_e^{Res_1}(x)) + x$. Described in Section A.3, by way of their construction, each of our selected edge geometric features is translation and rotation invariant to the network's input space. As discussed in Section A.5, we couple these features with our choice of node-wise positional encodings (see Section 4.2.3) to attain canonical invariant local frames for each residue to encode the relative poses of features in our protein graphs. In doing so, we leverage many of the benefits of employing equivariant representations while reducing the large memory requirements they typically induce, to yield a robust invariant representation of each input protein.

### 4.2.3 REMAINING TRANSFORMER INITIALIZATIONS AND OPERATIONS

For the initial node features used within the Geometric Transformer, we include each of DIPS-Plus' residue-level features described succinctly in Section 3. Additionally, we append initial min-max normalizations of each residue's index in $P_i$ to each node as node-wise positional encodings. For

the remainder of the Geometric Transformer's operations, the network's order of operations closely follows the definitions given by Dwivedi & Bresson (2021) for the Graph Transformer, with an exception being that the first normalization layer now precedes any affinity score calculations.

## 4.3 INTERACTION MODULE

Upon applying multiple layers of the Geometric Transformer to each pair of input protein chains, we then channel-wise interleave the Geometric Transformer's learned node representations $h_{\mathbb{A}}$ and $h_{\mathbb{B}}$ into $\mathbb{I}$ to serve as input to our interaction module, consisting of a dilated ResNet module adapted from Chen et al. (2021). The core residual network component in this interaction module consists of four residual blocks differing in the number of internal layers. Each residual block is comprised of several consecutive instance normalization layers and convolutional layers with 64 kernels of size $3 \times 3$. The number of layers in each block represents the number of 2D convolution layers in the corresponding component. The final values of the last convolutional layer are added to the output of a shortcut block, which is a convolutional layer with 64 kernels of size $1 \times 1$. A squeeze-and-excitation (SE) block (Hu et al. (2018)) is added at the end of each residual block to adaptively recalibrate its channel-wise feature responses. Ultimately, the output of the interaction module is a probability-valued $\mathbb{A}$ x $\mathbb{B}$ matrix that can be viewed as an inter-chain residue binding heatmap.

## 5 EXPERIMENTS

## 5.1 SETUP

For all experiments conducted with DeepInteract, we used 2 layers of the graph neural network chosen for the experiment and 128 intermediate GNN and CNN channels to restrict the time required to train each model. For the Geometric Transformer, we used an edge geometric neighborhood of size $n = 2$ for each edge such that each edge's geometric features are updated by their 4-nearest incoming edges. In addition, we used the Adam optimizer (Kingma & Ba (2014)), a learning rate of $1e^{-3}$, a weight decay rate of $1e^{-2}$, a dropout (i.e., forget) rate of 0.2, and a batch size of 1. We also employed 0.5-threshold gradient value clipping and stochastic weight averaging (Izmailov et al. (2018)). With an early-stopping patience period of 5 epochs, we observed most models converging after approximately 30 training epochs on DIPS-Plus. For our loss function, we used weighted cross entropy with a positive class weight of 5 to help the network overcome the large class imbalance present in interface prediction. All DeepInteract models employed 14 layers of our dilated ResNet architecture described in Section 4.3 and had their top-$k$ metrics averaged over three separate runs, each with a different random seed (standard deviation of top-$k$ metrics in parentheses). Prior to our experiments on the DB5 dataset's 55 test complexes, we fine-tuned each DeepInteract model using the held-out 140 and 35 complexes remaining in DB5 for training and validation, respectively. Employing a similar training configuration as described above, in this context we used a lower learning rate of $1e^{-5}$ to facilitate smoother transfer learning between DIPS-Plus and DB5.

## 5.2 HYPERPARAMETER SEARCH

To identify our optimal set of model hyperparameters, we performed a manual hyperparameter search over the ranges of $[1e^{-1}, 1e^{-2}, 1e^{-3}, 1e^{-4}, 1e^{-5}, 1e^{-6}]$ and $[1e^{-1}, 1e^{-2}, 1e^{-3}, 1e^{-4}]$ for the learning rate and weight decay rate, respectively. In doing so, we found a learning rate of $1e^{-3}$ and a weight decay rate of $1e^{-2}$ to provide the lowest loss and the highest metric values on our DIPS-Plus validation dataset. We restricted our hyperparameter search to the learning rate and weight decay rate of our models due to the large computational and environmental costs associated with training each model. However, this suggests further improvements to our models could be found with a more extensive hyperparameter search over, for example, the models' dropout rate.

## 5.3 SELECTION OF BASELINES

We considered the reproducibility and accessibility of a method to be the most important factors for its inclusion in our following benchmarks to encourage the adoption of accessible and transparent benchmarks for future works. As such, we have included the methods BIPSPI (an XGBoost-based algorithm) (Sanchez-Garcia et al. (2018)), DeepHomo (a CNN for homodimers) (Yan & Huang

Table 1: The average top-$k$ precision on two types of DIPS-Plus test targets.

| | 16 (Homo) | | | 16 (Hetero) | | |
|---|---|---|---|---|---|---|
| Method | 10 | $L/10$ | $L/5$ | 10 | $L/10$ | $L/5$ |
| BI | 0 | 0 | 0 | 0.02 | 0.02 | 0.02 |
| DH | 0.13 | 0.12 | 0.09 | | | |
| CC | | | | **0.17** | **0.16** | **0.15** |
| DI (GCN) | 0.22 (0.06) | 0.20 (0.07) | 0.18 (0.04) | 0.08 (0.01) | 0.08 (0.01) | 0.07 (0.02) |
| DI (GT) | 0.27 (0.06) | 0.24 (0.04) | 0.21 (0.04) | 0.10 (0.04) | 0.09 (0.04) | 0.08 (0.04) |
| DI (GeoT w/o EPE) | **0.28** (**0.05**) | 0.24 (0.01) | 0.23 (0.03) | 0.11 (0.05) | 0.10 (0.04) | 0.09 (0.03) |
| DI (GeoT w/o GFG) | 0.27 (0.08) | 0.24 (0.08) | 0.21 (0.08) | 0.10 (0.02) | 0.09 (0.02) | 0.09 (0.01) |
| DI (GeoT) | 0.25 (0.03) | **0.25** (**0.03**) | **0.23** (**0.02**) | 0.15 (0.04) | 0.14 (0.05) | 0.11 (0.04) |

Table 2: The average top-$k$ precision and recall on DIPS-Plus test targets of both types.

| | 32 (Both Types) | | | | | |
|---|---|---|---|---|---|---|
| Method | P@10 | P@$L/10$ | P@$L/5$ | R@$L$ | R@$L/2$ | R@$L/5$ |
| BI | 0.01 | 0.01 | 0.01 | 0.01 | 0.004 | 0.003 |
| DI (GCN) | 0.15 (0.03) | 0.16 (0.01) | 0.12 (0.02) | 0.10 (0.02) | 0.06 (0.01) | 0.03 (0.003) |
| DI (GT) | 0.18 (0.05) | 0.16 (0.04) | 0.15 (0.04) | 0.13 (0.02) | 0.07 (0.01) | 0.04 (0.01) |
| DI (GeoT w/o EPE) | 0.19 (0.04) | 0.18 (0.03) | 0.16 (0.03) | 0.14 (0.02) | 0.08 (0.02) | 0.04 (0.01) |
| DI (GeoT w/o GFG) | 0.18 (0.05) | 0.16 (0.04) | 0.15 (0.04) | 0.14 (0.02) | 0.08 (0.02) | 0.04 (0.01) |
| DI (GeoT) | **0.20** (**0.01**) | **0.19** (**0.01**) | **0.17** (**0.02**) | **0.15** (**0.003**) | **0.09** (**0.004**) | **0.04** (**0.002**) |

(2021)), and ComplexContact (a CNN for heterodimers) (Zeng et al. (2018)) since they are either easy to reproduce or simple for the general public to use to make predictions. Each method predicts interfacing residue pairs subject to the (on average) 1:1000 positive-negative class imbalance imposed by the biological sparsity of true interface contacts. We note that we also considered adding more recent baseline methods such as those of Townshend et al. (2019) and Liu et al. (2020). However, for both of these methods, we were not able to locate any provided source code or web server predictors facilitating the prediction of inter-protein residue-residue contacts for provided FASTA or PDB targets, so they ultimately did not meet our baseline selection criterion of reproducibility (e.g., an ability to make new predictions). We also include two ablation studies (e.g., DI (GeoT w/o GFG)) to showcase the effect of including network components unique to the Geometric Transformer.

Our selection criterion for each baseline method consequently determined the number of complexes against which we could feasibly test each method, thereby restricting the size of our test datasets to 106 complexes in total. In addition, not all baselines chosen were originally trained for both types of protein complexes (i.e., homodimers and heterodimers), so for these baselines we do not include their results for the type of complex for which they are not respectively designed.

For brevity, in all experiments, we refer to BIPSPI, DeepHomo, ComplexContact, and DeepInteract as BI, DH, CC, and DI, respectively. Further, we refer to the Graph Convolutional Network of Kipf & Welling (2016), the Graph Transformer of Dwivedi & Bresson (2021), and the Geometric Transformer as GCN, GT, and GeoT, respectively. To assess models' ability to correctly select residue pairs in interaction upon binding of two given chains, all methods are scored using the top-$k$ precision and recall metrics (defined in Section A.2) commonly used for intra-chain contact prediction (Chen et al. (2021)) as well as recommender systems (Jiang et al. (2020)), where $k \in \{10, L/10, L/5, L/2\}$ with $L$ being the length of the shortest chain in a given complex.

### 5.4 DISCUSSION

Table 1 demonstrates that DeepInteract outperforms or achieves competitive results compared to existing state-of-the-art methods for interface contact prediction on DIPS-Plus with both types of protein complexes, homodimers (homo) where the two chains are of the same protein and heterodimers (hetero) where the two chains are of different proteins. Table 2 shows that, when taking both types of complexes into account, DeepInteract outperforms all other methods' predictions on DIPS-Plus. Since future users of DeepInteract may want to predict interface contacts for either type of complex, we consider a method's type-averaged top-$k$ metrics as important metrics for which to optimize.

Likewise, Tables 3 and 4 present the average top-$k$ metrics of DeepInteract on 19 challenging protein complexes (14 homodimers and 5 heterodimers) from the 13th and 14th rounds of the joint CASP-CAPRI meeting. In them, we once again see DeepInteract exceed the precision of state-of-the-art in-

Table 3: The average top-$k$ precision on dimers from CASP-CAPRI 13 & 14.

| Method | 14 (Homo) | | | 5 (Hetero) | | |
|---|---|---|---|---|---|---|
| | 10 | $L/10$ | $L/5$ | 10 | $L/10$ | $L/5$ |
| BI | 0 | 0 | 0 | 0.04 | 0 | 0.03 |
| DH | 0.02 | 0.02 | 0.02 | | | |
| CC | | | | 0.06 | 0.08 | 0.05 |
| DI (GCN) | 0.12 (0.04) | 0.11 (0.03) | **0.13 (0.02)** | 0.10 (0.07) | 0.11 (0.08) | 0.09 (0.04) |
| DI (GT) | 0.08 (0.03) | 0.09 (0.05) | 0.08 (0.03) | 0.14 (0.02) | 0.14 (0.02) | 0.12 (0.03) |
| DI (GeoT w/o EPE) | 0.11 (0.01) | 0.12 (0.02) | 0.11 (0.01) | 0.18 (0.07) | 0.20 (0.09) | 0.18 (0.04) |
| DI (GeoT w/o GFG) | 0.10 (0.02) | 0.10 (0.02) | 0.09 (0.02) | 0.14 (0.03) | 0.17 (0.03) | 0.14 (0.02) |
| DI (GeoT) | **0.18 (0.05)** | **0.13 (0.03)** | 0.11 (0.02) | **0.30 (0.09)** | **0.31 (0.07)** | **0.24 (0.04)** |

Table 4: The average top-$k$ precision and recall across all targets from CASP-CAPRI 13 & 14.

| Method | 19 (Both Types) | | | | | |
|---|---|---|---|---|---|---|
| | P@10 | P@$L/10$ | P@$L/5$ | R@$L$ | R@$L/2$ | R@$L/5$ |
| BI | 0.01 | 0 | 0.01 | 0.02 | 0.01 | 0.001 |
| DI (GCN) | 0.12 (0.04) | 0.10 (0.05) | 0.09 (0.04) | 0.11 (0.001) | 0.06 (0.01) | 0.02 (0.01) |
| DI (GT) | 0.10 (0.03) | 0.09 (0.03) | 0.08 (0.02) | 0.11 (0.02) | 0.06 (0.01) | 0.02 (0.01) |
| DI (GeoT w/o EPE) | 0.13 (0.02) | 0.14 (0.03) | 0.13 (0.02) | 0.12 (0.01) | 0.07 (0.01) | 0.03 (0.01) |
| DI (GeoT w/o GFG) | 0.11 (0.01) | 0.12 (0.02) | 0.10 (0.02) | 0.11 (0.01) | 0.06 (0.01) | 0.03 (0.01) |
| DI (GeoT) | **0.21 (0.01)** | **0.19 (0.01)** | **0.14 (0.01)** | **0.13 (0.02)** | **0.08 (0.01)** | **0.04 (0.003)** |

Table 5: The average top-$k$ precision and recall on DB5 test targets.

| Method | 55 (Hetero) | | | | | |
|---|---|---|---|---|---|---|
| | P@10 | P@$L/10$ | P@$L/5$ | R@$L$ | R@$L/2$ | R@$L/5$ |
| BI | 0 | 0.002 | 0.001 | 0.003 | 0.001 | 0.0004 |
| CC | 0.002 | 0.003 | 0.003 | 0.007 | 0.003 | 0.001 |
| DI (GCN) | 0.005 (0.002) | 0.006 (0.001) | 0.007 (0.001) | 0.013 (0.002) | 0.008 (0.001) | 0.003 (0.001) |
| DI (GT) | 0.008 (0.004) | 0.008 (0.005) | 0.008 (0.004) | 0.010 (0.005) | 0.006 (0.003) | 0.003 (0.002) |
| DI (GeoT w/o EPE) | 0.011 (0.004) | 0.009 (0.004) | 0.011 (0.002) | 0.018 (0.01) | 0.010 (0.004) | 0.0034 (0.002) |
| DI (GeoT w/o GFG) | 0.008 (0.001) | 0.008 (0.001) | 0.009 (0.002) | 0.014 (0.01) | 0.006 (0.002) | 0.003 (0.001) |
| DI (GeoT) | **0.013 (0.001)** | **0.009 (0.003)** | **0.011 (0.001)** | **0.018 (0.001)** | **0.010 (0.001)** | **0.0034 (0.001)** |

terface contact predictors for both complex types. In particular, we see that combining DeepInteract with the Geometric Transformer offers improvements to the majority of our top-$k$ metrics for both homodimers and heterodimers compared to using either a GCN or a Graph Transformer-based GNN backbone, notably for heteromeric complexes with largely asymmetric inter-chain geometries. Such a result supports our hypothesis that the Geometric Transformer's geometric self-attention mechanism can enable enhanced prediction performance for downstream tasks on geometrically-intricate 3D objects such as protein structures, using interface contact prediction as a case study.

Finally, in Table 5, we observe that, in predicting the interface contacts between *unbound* protein chains in the DB5 test dataset, the Geometric Transformer enables enhanced top-$k$ precision and recall (definition in A.2) compared to all other baseline methods, including GCNs and Graph Transformers paired with DeepInteract. Such as result confirms, to a degree, the Geometric Transformer's ability to predict how the structural conformations occurring upon the binding of two protein chains influence which inter-chain residue pairs will interact with one another in the complex's *bound* state.

## 6 CONCLUSION

We presented DeepInteract which introduces the geometry-evolving Geometric Transformer for protein structures and demonstrates its effectiveness in predicting residue-residue interactions in protein complexes. We foresee several other uses of the Geometric Transformer in protein deep learning such as quaternary structure quality assessment and residue disorder prediction, to name a few. One limitation of the Geometric Transformer's current design is the high computational complexity associated with its dot product self-attention mechanism, which we hope to overcome using efficient alternatives to self-attention like that of the Nyströmformer (Xiong et al. (2021)).

## ACKNOWLEDGMENTS

The project is partially supported by two NSF grants (DBI 1759934 and IIS 1763246), one NIH grant (GM093123), three DOE grants (DE-SC0020400, DE-AR0001213, and DE-SC0021303), and the computing allocation on the Summit compute cluster provided by Oak Ridge Leadership Computing Facility (Contract No. DE-AC05-00OR22725).

## ETHICS STATEMENT

DeepInteract is designed to be used for machine learning of protein molecular data. It makes use of only publicly available information concerning biomolecular structures and their interactions. Consequently, all data used to create DeepInteract models do not contain any personally identifiable information or offensive content. As such, we do not foresee negative societal impacts as a consequence of DeepInteract or the Geometric Transformer being made publicly available. Furthermore, future adaptions or enhancements to DeepInteract may benefit the machine learning community and, more broadly, the scientific community by providing meaningful refinements to a transparent and extensible pipeline for geometric deep learning of protein-protein interactions.

## REPRODUCIBILITY STATEMENT

To enable this work to be reproducible by others, we have thoroughly documented two methods for running predictions with DeepInteract using a provided PyTorch Lightning model checkpoint (Falcon (2019)) and one method for training DeepInteract models in an associated GitHub repository. The most convenient prediction method describes how to run DeepInteract as a platform-agnostic Docker container. The second method, for both training and prediction, details how users can piece-wise install DeepInteract and its data and software dependencies on a Linux-based operating system.

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

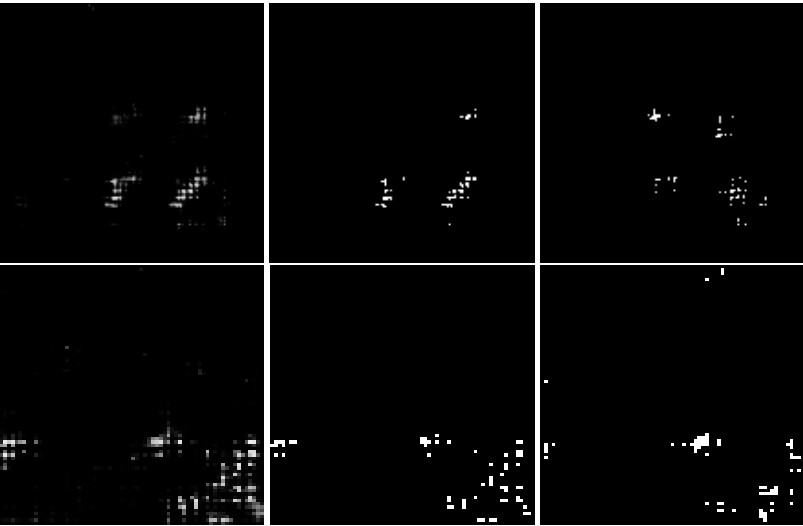

Figure 5: The network's softmax contact probabilities (leftmost column), 0.5 positive probability-thresholded predictions (middle column), and ground-truth labels (rightmost column), respectively, for PDB ID: 4HEQ (first row) and 6TRI (second row), two of the complexes in our test datasets.

# A   APPENDIX

## A.1   SAMPLE INTERFACE CONTACT PREDICTIONS

In the first row of Figure 5, we see predictions made by DeepInteract for a homodimer complex from our test partition of DIPS-Plus (i.e., PDB ID: 4HEQ). The leftmost image represents the softmax contact probability map. The center image corresponds to the same contact map after having a 0.5 probability threshold applied to it such that residue pairs with at least a 50% probability of being in interaction with each other have their interaction probabilities rounded up to 1.0. The rightmost image is the ground-truth contact map. Similarly, in the second row of Figure 5, we are shown the cropped predictions made by DeepInteract for a CASP-CAPRI test heterodimer (i.e., PDB ID: 6TRI).

## A.2   TOP-K TEST PRECISION AND RECALL OF BOTH COMPLEX TYPES IN DIPS-PLUS AND CASP-CAPRI

Formally, our definitions of a model's top-$k$ precision $prec_k$ and recall $rec_k$, where $T_{pos_k}$ represents the number of true positive residue pairs selected from a model's top-$k$ most probable pairs and $T_{pos}$ corresponds to the total number of true positive pairs in the complex, are

$$prec_k = \frac{T_{pos_k}}{k} \tag{6}$$

and

$$rec_k = \frac{T_{pos_k}}{T_{pos}}. \tag{7}$$

After defining top-$k$ recall as such, in Tables 6 and 7 we provide the results of each model's top-$k$ recall in the same set of experiments as given in Section 5.

## A.3   DEFINITION OF EDGE GEOMETRIC FEATURES

Similar to Ingraham et al. (2019), we construct a local reference frame (i.e., an orientation $\boldsymbol{O}_i$) for each protein chain graph's residues. Representing each residue by its Cartesian coordinates $\boldsymbol{x}_i$, we formally define

Table 6: The average top-$k$ recall on two types of DIPS-Plus test targets.

| Method | 16 (Homo) | | | 16 (Hetero) | | |
| --- | --- | --- | --- | --- | --- | --- |
| | R@$L$ | R@$L/2$ | R@$L/5$ | R@$L$ | R@$L/2$ | R@$L/5$ |
| BI | 0.01 | 0 | 0 | 0.01 | 0.01 | 0.01 |
| DH | 0.07 | 0.04 | 0.02 | | | |
| CC | | | | **0.17** | **0.12** | **0.07** |
| DI (GCN) | 0.14 (0.03) | 0.08 (0.01) | 0.04 (0.01) | 0.08 (0.02) | 0.05 (0.02) | 0.02 (0.01) |
| DI (GT) | 0.17 (0.01) | 0.10 (0.01) | 0.05 (0.01) | 0.09 (0.02) | 0.05 (0.02) | 0.03 (0.01) |
| DI (GeoT w/o EPE) | 0.18 (0.02) | 0.11 (0.01) | 0.05 (0.01) | 0.11 (0.03) | 0.07 (0.02) | 0.03 (0.02) |
| DI (GeoT w/o GFG) | 0.19 (0.04) | 0.11 (0.03) | 0.05 (0.02) | 0.09 (0.01) | 0.05 (0.02) | 0.03 (0.01) |
| DI (GeoT) | **0.19 (0.004)** | **0.12 (0.004)** | **0.06 (0.003)** | 0.12 (0.003) | 0.07 (0.01) | 0.03 (0.01) |

Table 7: The average top-$k$ recall on dimers from CASP-CAPRI 13 & 14.

| Method | 14 (Homo) | | | 5 (Hetero) | | |
| --- | --- | --- | --- | --- | --- | --- |
| | R@$L$ | R@$L/2$ | R@$L/5$ | R@$L$ | R@$L/2$ | R@$L/5$ |
| BI | 0.02 | 0.01 | 0 | 0.01 | 0 | 0 |
| DH | 0.02 | 0.01 | 0 | | | |
| CC | | | | 0.03 | 0.01 | 0.01 |
| DI (GCN) | 0.10 (0.01) | 0.07 (0.01) | 0.04 (0.02) | 0.08 (0.04) | 0.04 (0.02) | 0.02 (0.01) |
| DI (GT) | 0.10 (0.01) | 0.06 (0.01) | 0.02 (0.01) | 0.10 (0.01) | 0.05 (0.01) | 0.02 (0.01) |
| DI (GeoT w/o EPE) | 0.11 (0.01) | 0.07 (0.01) | 0.04 (0.01) | 0.12 (0.02) | 0.07 (0.01) | 0.03 (0.01) |
| DI (GeoT w/o GFG) | 0.10 (0.02) | 0.06 (0.01) | 0.03 (0.01) | 0.11 (0.02) | 0.07 (0.01) | 0.03 (0.01) |
| DI (GeoT) | **0.12 (0.03)** | **0.07 (0.01)** | **0.04 (0.01)** | **0.15 (0.02)** | **0.09 (0.01)** | **0.04 (0.01)** |

$$\boldsymbol{u}_i = \frac{\boldsymbol{x}_i - \boldsymbol{x}_{i-1}}{\|\boldsymbol{x}_i - \boldsymbol{x}_{i-1}\|}, \quad \boldsymbol{n}_i = \frac{\boldsymbol{u}_i \times \boldsymbol{u}_{i+1}}{\|\boldsymbol{u}_i \times \boldsymbol{u}_{i+1}\|}, \quad \boldsymbol{b}_i = \frac{\boldsymbol{u}_i - \boldsymbol{u}_{i+1}}{\|\boldsymbol{u}_i - \boldsymbol{u}_{i+1}\|}. \tag{8}$$

with $\boldsymbol{n}_i$ being the unit vector normal to the plane formed by the rays $(\boldsymbol{x}_{i-1} - \boldsymbol{x}_i)$ and $(\boldsymbol{x}_{i+1} - \boldsymbol{x}_i)$ and $\boldsymbol{b}_i$ being the negative bisector of this plane. We then define $\boldsymbol{O}_i$ as

$$\boldsymbol{O}_i = [\boldsymbol{b}_i \;\; \boldsymbol{n}_i \;\; \boldsymbol{b}_i \times \boldsymbol{n}_i]. \tag{9}$$

Having defined the orientation $\boldsymbol{O}_i$ for each residue that describes the local reference frame $(\boldsymbol{x}_i, \boldsymbol{O}_i)$. To provide the Geometric Transformer with an alternative notion of residue-residue orientations, we define the unit vector normal to the amide plane for residue $i$ as

$$\boldsymbol{U}_i = (\boldsymbol{x}_{C\alpha_i} - \boldsymbol{x}_{C\beta i}) \times (\boldsymbol{x}_{C\beta_i} - \boldsymbol{x}_{N_i}) \tag{10}$$

where $\boldsymbol{x}_{C\alpha_i}$, $\boldsymbol{x}_{C\beta i}$, and $\boldsymbol{x}_{Ni}$ are the Cartesian coordinates of the residue's carbon-alpha ($C\alpha$), carbon-beta ($C\beta$), and nitrogen ($N$) atoms, respectively.

Finally, we relate the reference frames for residues $i$ and $j$ by describing their edge geometric features as

$$\left( \boldsymbol{r}(\|\boldsymbol{x}_j - \boldsymbol{x}_i\|), \;\; \boldsymbol{O}_i^T \frac{\boldsymbol{x}_j - \boldsymbol{x}_i}{\|\boldsymbol{x}_j - \boldsymbol{x}_i\|}, \;\; \boldsymbol{q}(\boldsymbol{O}_i^T \boldsymbol{O}_j), \;\; \boldsymbol{a}(\boldsymbol{U}_i, \boldsymbol{U}_j) \right) \tag{11}$$

with the first term $\boldsymbol{r}()$ being a distance encoding of 16 Gaussian RBFs spaced isotropically from 0 to 20 Å, the second term describing the relative direction of $\boldsymbol{x}_j$ with respect to reference frame $(\boldsymbol{x}_i, \boldsymbol{O}_i)$, the third term detailing an orientation encoding $\boldsymbol{q}()$ of the quaternion representation of the rotation matrix $\boldsymbol{O}_i^T \boldsymbol{O}_j$, representing each quaternion with respect to its vector of real coefficients, and the fourth term $\boldsymbol{a}()$ representing the angle between the amide plane normal vectors $\boldsymbol{U}_i$ and $\boldsymbol{U}_j$.

Our definition of these edge geometric features makes use of the backbone atoms for each residue. As such, the graph representation of protein chains we use with the Geometric Transformer encodes not only residue-level geometric features but also those derived from an atomic view of protein structures. We hypothesized this hybrid approach to modeling protein structure geometries would have a noticeable downstream effect on interface contact prediction precision via the node and edge representations learned by the Geometric Transformer. This hypothesis is confirmed in Section 5.4.

Table 8: The protein complexes selected from DIPS-Plus for testing interface contact predictors.

| PDB ID | Chain 1 | Chain 2 | Type | PDB ID | Chain 1 | Chain 2 | Type |
|--------|---------|---------|------|--------|---------|---------|------|
| 1BHN | B | D | Homo | 1AON | R | S | Hetero |
| 1KPT | A | B | Homo | 1BE3 | D | E | Hetero |
| 1SDU | A | B | Homo | 1GK8 | K | M | Hetero |
| 1UZN | A | B | Homo | 1OCZ | R | V | Hetero |
| 2B4H | A | B | Homo | 1UWA | A | I | Hetero |
| 2G30 | C | E | Homo | 3A6N | A | E | Hetero |
| 2GLM | E | F | Homo | 3ABM | D | K | Hetero |
| 2IUO | D | J | Homo | 3JRM | H | I | Hetero |
| 3BXS | A | B | Homo | 3MG6 | D | E | Hetero |
| 3CT7 | B | E | Homo | 3MNN | C | F | Hetero |
| 3NUT | A | D | Homo | 3T1Y | E | H | Hetero |
| 3RE3 | B | C | Homo | 3TUY | D | E | Hetero |
| 4HEQ | A | B | Homo | 3VYG | G | H | Hetero |
| 4LIW | A | B | Homo | 4A3D | C | L | Hetero |
| 4OTA | D | F | Homo | 4CW7 | G | H | Hetero |
| 4TO9 | B | D | Homo | 4DR5 | G | I | Hetero |

Table 9: The CASP-CAPRI 13-14 protein complexes selected for testing interface contact predictors.

| PDB ID | Chain 1 | Chain 2 | Type |
|--------|---------|---------|------|
| 5W6L | A | B | Homo |
| 6D2V | A | B | Homo |
| 6E4B | A | B | Homo |
| 6FXA | C | D | Homo |
| 6HRH | A | B | Homo |
| 6MXV | A | B | Homo |
| 6N64 | A | B | Homo |
| 6N91 | A | B | Homo |
| 6NQ1 | A | B | Homo |
| 6QEK | A | B | Homo |
| 6UBL | A | B | Homo |
| 6UK5 | A | B | Homo |
| 6YA2 | A | B | Homo |
| 7CWP | C | D | Homo |
| 6CP8 | A | C | Hetero |
| 6D7Y | A | B | Hetero |
| 6TRI | A | B | Hetero |
| 6XOD | A | B | Hetero |
| 7M5F | A | C | Hetero |

## A.4 PROTEIN COMPLEXES SELECTED FOR TESTING

To facilitate reproducibility of the results presented in Section 5.4, Table 8 displays the PDB and chain IDs of DIPS-Plus protein complexes chosen for testing. Likewise, in Table 9, we provide the PDB and chain IDs of CASP-CAPRI 13-14 targets chosen for testing. These two tables describe precisely which targets were selected and ultimately used in our RCSB-derived benchmarks. For full data provenance, the targets we selected from the Docking Benchmark 5 dataset (Vreven et al. (2015)) for benchmarking are the same 55 protein heterodimers used for testing in works such as that of Fout et al. (2017), Townshend et al. (2019), and Liu et al. (2020).

## A.5 Invariance or Equivariance?

In our view, a natural question to ask concerning a deep learning architecture designed for a specific task is whether equivariance to translations and rotations in $\mathbb{R}^3$ should be preferred over invariance to transformations in such a geometric space. The benefits of employing equivariant representations in a deep learning architecture primarily include symmetry-preserving updates to type-1 tensors such as the coordinates representing an object in $\mathbb{R}^3$ and the derivation of invariant relative feature poses for type-0 features such as scalars (Cohen & Welling (2016)). However, equivariant representations, particularly those derived with a self-attention mechanism, typically induce large memory requirements for training and inference. In contrast, in the context of data domains such as ordered sets or proteins where there exists a canonical ordering of points, invariant representations may be adopted to simultaneously reduce memory requirements and provide many of the benefits of using equivariant representations such as attaining these relative poses of type-0 features (Ingraham et al. (2019) and Jumper et al. (2021)). As such, in the context of the Geometric Transformer, we decided to pursue invariance over equivariance, to reduce the network's effective memory requirements and to improve its learning efficiency and generalization capabilities (Bronstein et al. (2021)). However, for applications such as protein-protein docking that may more directly rely on type-1 tensors for network predictions (Costa et al. (2021)), designing one's network architecture to preserve full translation and rotation equivariance in $\mathbb{R}^3$ is, in our perspective, a worthwhile research direction to pursue as many promising results on molecular datasets have already been demonstrated with equivariant neural networks such as SE(3)-Transformers (Fuchs et al. (2020)) and lightweight graph architectures such as the Equivariant Graph Neural Network (Satorras et al. (2021)).

## A.6 Rationale behind the Node Initialization Scheme

DIPS-Plus residue-level features are initially embedded in our protein chain graphs to accelerate the network's training. However, we also initially append node-wise min-max positional encodings in our network's operations. We do this to initialize the Geometric Transformer with information concerning the residue ordering of the chain's underlying sequence as such ordering is important to understanding downstream protein structural, interactional, and functional properties of each residue.

## A.7 Rationale behind the Edge Initialization Module's Design

For the edge initializer module's four protein geometric features, we sought to include enough geometric information for the network to be able to uniquely determine the Euclidean positions of each node's neighboring nodes. For this reason, we adopt similar distance, direction, and orientation descriptors as Ingraham et al. (2019). We concatenate the protein backbone-geometric features provided by inter-residue distances, directions, and orientations with the angles between each residue pair's amide plane normal vectors. This is done ultimately to apply gating to each edges' messages, distances, directions, orientations, and amide angles separately to encourage the network to learn the importance of specific channels in each of these input features. Gating is a technique that has previously been shown to encourage neural networks to not become over-reliant on any particular input feature (Gu et al. (2020)) and, as such, in the Geometric Transformer can be seen as a form of channel-wise dropout for single feature sets. By also employing residual connections from original edge representations to gating-learned edge representations, the network module can operate more stably in the presence of multiple neural network layers (He et al. (2016)). Furthermore, in the edge initialization module, we introduce edge-wise sinusoidal position encodings to provide the network with a directional notion of residue-to-residue distances in protein chains' underlying sequences.

## A.8 Rationale behind the Conformation Module's Design

The conformation module's design was inspired by SphereNet (Liu et al. (2021b)) and similar graph neural network architectures designed for learning on 3D graphs. What distinguishes our conformation module from the works of others is its introduction of the notion of $2n$ edge geometric neighborhoods when updating edge representations as well as its incorporation of geometric insights specific to large biomolecules such as proteins. Namely, by including the residue-residue distances, residue-residue local reference frame directions and (quaternion) orientations, and amide plane-amide plane angles, the network is provided with enough information to ascertain the rela-

tive coordinates of each neighboring residue from a given residue's local reference frame (Liu et al. (2021b)), thereby ensuring the network's capability of adequately learning from 3D structures.

## A.9   ALTERNATIVE NETWORKS WITHIN THE INTERACTION MODULE

We, like Liu et al. (2020), note that the task of interface prediction bears striking similarities to dense prediction tasks in computer vision (e.g., semantic segmentation). In this train of thought, we experimented with several semantic segmentation models as replacements for our interaction module's dilated ResNet, one namely being DeepLabV3Plus (Chen et al. (2018)). We observed a strong propensity of such semantic segmentation models to identify interaction regions well but to do so with low *pixel-wise* precision. We hypothesize this is due to the downsampling and up-sampling methods often employed within such architectures that invariably degrade the original input tensor's representation resolution. We also experimented with several state-of-the-art Vision Transformer and MLP-based models for computer vision but ultimately found their algorithmic complexity, memory usage, or input shape requirements to be prohibitive for this task, since our test datasets' input protein complexes can vary greatly in size to contain between 20 residues and over 2,000 residues in length. As such, for the design of DeepInteract's interaction module, we experimented primarily with convolution-based architectures that do not employ such sampling techniques or pose limited input size constraints.

## A.10   HARDWARE USED

The Oak Ridge Leadership Facility (OLCF) at the Oak Ridge National Laboratory (ORNL) is an open science computing facility that supports HPC research. The OLCF houses the Summit compute cluster. Summit, launched in 2018, delivers 8 times the computational performance of Titan's 18,688 nodes, using only 4,608 nodes. Like Titan, Summit has a hybrid architecture, and each node contains multiple IBM POWER9 CPUs and NVIDIA Volta GPUs all connected with NVIDIA's high-speed NVLink. Each node has over half a terabyte of coherent memory (high bandwidth memory + DDR4) addressable by all CPUs and GPUs plus 800GB of non-volatile RAM that can be used as a burst buffer or as extended memory. To provide a high rate of I/O throughput, the nodes are connected in a non-blocking fat-tree using a dual-rail Mellanox EDR InfiniBand interconnect. We used the Summit compute cluster to train all our models.

## A.11   SOFTWARE USED

In addition, we used Python 3.8 (Van Rossum & Drake (2009)), PyTorch 1.7.1 (Paszke et al. (2019)), and PyTorch Lightning 1.4.8 (Falcon (2019)) to run our deep learning experiments. PyTorch Lightning was used to facilitate model checkpointing, metrics reporting, and distributed data parallelism across 72 Tesla V100 GPUs. A more in-depth description of the software environment used to train and predict with DeepInteract models can be found at https://github.com/BioinfoMachineLearning/DeepInteract.

