# OpenReview forum: "Geometric Transformers for Protein Interface Contact Prediction"
_ICLR.cc/2022/Conference — ICLR 2022 Poster_

### Official Review · Reviewer_rPiF · 2021-10-28

**Correctness:** 4
**Technical Novelty And Significance:** 2
**Empirical Novelty And Significance:** 3
**Recommendation:** 6
**Confidence:** 5

**Main Review:**

##########################################################################

Pros:

- The paper provided the first example of graph self-attention applied to protein interface contact prediction, which is an interesting idea for the protein bioinformatics area.
- The proposed Geometric Transformer can be used for tasks on 3D protein structures and other 3D graphs. The author trains the Geometric Transformer to evolve a geometric representation of protein structures simultaneously with protein sequence and coevolutionary features for the prediction of interchain residue-residue contacts, demonstrating the merit of the recently released Enhanced Database of Interacting Protein Structures (DIPS-Plus) for interface prediction.
- The paper is well-written and the design decisions are clearly explained. The comparison of benchmark methods is also interesting to read.

##########################################################################

Cons:

- The core idea of this paper is Geometric Transformer, however, it seems that the difference between Geometric Transformer and Graph Transformer is not very big except for the last layer, because the essence of these two models is multi-head attention. Although the author stated in the paper the new modules of Geometric Transformer, including EDGE REPRESENTATION INITIALIZATION, CONFORMATION MODULE, NODE REPRESENTATION INITIALIZATION, and INTERACTION MODULE, it will be more convincing if the author can compare with Graph Transformer.
- In the model description, the author lacks the statement of the objective function and loss function.
- In the experiment comparison, if the author can add an ablation study experiment, it will be more convincing. In addition, from the experimental results in Table 1 and Table 2, Geometric Transformer does not seem to be significantly improved compared to Graph Transformer. Can the author provide more explanations?
- From Figure 1, the author provided the visualization of interacting protein chains, can the author also provide visualization of the predicted results? Such as qualitative analysis.

**Summary Of The Paper:**

This paper proposed a novel geometry-evolving graph transformer for protein interface contact prediction, named DeepInteract. In particular, it predicts partner-specific protein interface contacts (i.e., inter-protein residue-residue contacts) given the 3D tertiary structures of two proteins as input, showcasing its effective use in learning representations of protein geometries to be exploited in downstream tasks. The experiments on challenging protein complex targets also demonstrated the proposed method achieves SOTA results for interface contact prediction.

**Summary Of The Review:**

Considering the above pros and cons, my recommendation of the paper is marginally above the acceptance threshold.

---

> ### Author Response · Authors · 2021-11-19
> **Response to Feedback from Reviewer rPiF (1)**
>
> We authors would like to thank you for carefully reading and providing feedback on our manuscript. We have recently uploaded the latest version of our manuscript based on each reviewer's feedback. Above, we have summarized our main changes to the manuscript since its initial submission for review. However, we would like to point out here some specific changes we made based on your review. The following present our most relevant changes concerning your points of feedback.
>
> ---
> Latest Changes to the Manuscript:
>
> 1. We have rewritten Section 4 (Methods) in the manuscript to (1) more clearly highlight the novelties of our new Geometric Transformer compared to the Graph Transformer or other Transformer-like architectures. One way we have done this is by moving much of our discussion of the network modules' design rationales to our manuscript's supplementary material (i.e., Section A - Appendix), to make room in the main text for 5 key new equations describing the operations of the Geometric Transformer's $\textit{Edge}$ $\textit{Initialization}$ $\textit{Module}$ and geometry-evolving $\textit{Conformation}$ $\textit{Module}$. We note that two of these equations describe what we consider to be a primary contribution of our manuscript: the definition and usage of edge geometric neighborhoods in updating geometric representations of 3D graphs. Taken directly from our revised Section 4.2.2 (Conformation Module), the following equations (i.e., Equations 3-5), in our view, succinctly summarize some of the major contributions of our work, demonstrating how we define the edge geometric neighborhood of an edge $e$ to be the $2n$ edges
>
> \\[
> \mathbb{E}\_{k} = \\{e\_{n\_{1}i}, e\_{n\_{2}j} \mid (n\_{1}, n\_{2} \in \mathbb{N}\_{k})\ \mathrm{and}\ (n\_{1}, n\_{2} \ne i, j)\\},\ \ \ \ \ (3)
> \\]
>
> where $\mathbb{N}\_{k} \subset \mathbb{N}$ are the source nodes for incoming edges on edge $e's$ source and destination nodes. We then give one of several possible ways in which one can use this definition of an edge geometric neighborhood to update the edge representation $e_{ij}$ of edge $e$:
>
> \\[
> O\_{ij}=\sum\_{k \in \mathbb{E}\_{k}} \ [(\phi\_{e}^{n}(e\_{ij, k}^{n}) \odot \phi\_{e}^{\mathrm{f}\_{n}}(\mathrm{f}\_{n})), \forall n \in \mathbb{F}]\ \ \ \ \ (4)
> \\]
>
> \\[
> e\_{ij} = 2 \times ResBlock\_{2}(\phi\_{e}^{5}(e\_{ij}) + 2 \times ResBlock\_{1}(\phi\_{e}^{5}(e\_{ij}) + O\_{ij})),\ \ \ \ \ (5)
> \\]
>
> where each term referenced above is defined correspondingly in Section 4.2.2 of our latest manuscript.
>
> To the best of our knowledge, no previous works have considered using the nearest neighborhoods of edges behaving as pseudo-nodes to update the representation of each edge. As such, we have outlined a blueprint from which others can investigate new techniques for using similar edge neighborhoods in future network architectures.
>
> 2. Thank you for mentioning this point. In Section 5.1 (Experiments - Setup) of our latest version of the manuscript, we specify class-weighted cross entropy loss as our models' loss function. If you believe this detail would best fit in the model description instead, we would be happy to change its placement.
>
> 3. To give readers more certainty about the performance and stability of our proposed method, we have now included experiments on the Docking Benchmark 5 dataset of unbound protein chains to allow others to easily port their future results for comparison since this dataset has traditionally been used to assess the performance of interface contact predictors. We have also included the top-k recall metric to complement our experiments' top-k precision scores, to further elucidate the behavior of our models on each protein complex dataset we use for benchmarking. Regarding the Geometric Transformer's performance compared to that of the Graph Transformer and the Graph Convolutional Network, in our summary response above to all reviewers, we give one possible explanation for why, in limited cases, one of these models may surpass the Geometric Transformer's performance on homodimers by a small margin. In short, we hypothesize this may be due to the Geometric Transformer prioritizing learning protein targets with more complex inter and intra-chain geometries (e.g., heterodimeric proteins which are often considered by bioinformaticians to be harder targets against which to predict). This hypothesis is supported by all three experiments we conducted in our manuscript, on the DIPS-Plus, CASP-CAPRI, and DB5 datasets, respectively.
>
> 4. In our supplementary material (Appendix - A.1), we provide sample prediction images for readers.
>
> ---
> We would like to thank you in advance for any additional feedback you may have for our manuscript in its current state. Thank you once again for your time and constructive comments!

---

> > ### Comment · Reviewer_rPiF · 2021-11-30
> > **Respond to the authors' feedback**
> >
> > I have read the authors' feedback and really appreciate the authors' so detailed reply and update, good job!

---

> > > ### Author Response · Authors · 2021-11-30
> > > **Update to Equations and Addition of Ablation Studies**
> > >
> > > Dear Reviewer rPiF,
> > >
> > > Thank you very much for your kind remarks.
> > > We thought it worth mentioning that your comment reminded us to update our original response to your initial review. Specifically, it previously displayed an older (unrevised) version of Equations 3, 4, and 5 from our paper. We have since updated these three equations in our original response above (Response to Feedback from Reviewer rPiF (1)) to match their latest versions in our local copy of the manuscript.
> > >
> > > Additionally, we have since added two ablation studies (https://openreview.net/forum?id=CS4463zx6Hi&noteId=K9x9kFPMxXT and https://openreview.net/forum?id=CS4463zx6Hi&noteId=8FFykBSMbmd) experimentally confirming the superiority of the Geometric Transformer for protein structures compared to the original Graph Transformer.
> > >
> > > Best,
> > >
> > > Paper 3368 Authors

---

> ### Author Response · Authors · 2021-11-19
> **Response to Feedback from Reviewer rPiF (2)**
>
> Continuing from our previous list of responses, we would like to mention that, given the restriction on the amount of time remaining for us to make changes to our manuscript, we would posit that, in lieu of having a 3D visualization of model predictions in the main text, readers will still be able to view 2D images of predicted contact probability maps (i.e., inter-chain heatmaps) in the supplementary material. If we are, in fact, able to add such a 3D visualization of predictions to the main text before the revision deadline on Monday, we would certainly be delighted to make a meaningful addition like this, thanks to your suggestion.

---

### Official Review · Reviewer_Qnz5 · 2021-11-01

**Correctness:** 3
**Technical Novelty And Significance:** 2
**Empirical Novelty And Significance:** 1
**Recommendation:** 5
**Confidence:** 2

**Main Review:**


Weakness:

1. The key contribution of this paper is not clear. The author should clarify the difference between the proposed method and the graph transformer (Dwivedi & Bresson (2021). Several related works should also be discussed, such as [1] and [2].

2. The method section as well as the figures is hard to read. The author should use equations to describe the computations of the DL model.

3. More than two important baselines (such as [2] and [3]) are not included. The author should also conduct the experiments on the DB5 dataset, which is widely used for protein-protein contact prediction.


[1] Kristof T Schütt, Farhad Arbabzadah, Stefan Chmiela, Klaus R Müller, and Alexandre Tkatchenko. Quantum-chemical insights from deep tensor neural networks. Nature communications, 8:13890, 2017.

[2] Liu, X., Luo, Y., Li, P., Song, S., & Peng, J. (2021). Deep geometric representations for modeling effects of mutations on protein-protein binding affinity. PLoS computational biology, 17(8), e1009284.

[3] Yi Liu et al. “Deep learning of high-order interactions for protein interface prediction”. In: Proceedings of the 26th ACM SIGKDD International Conference on Knowledge Discovery & Data Mining. 2020, pp. 679–687.

Overall, I am inclined to reject. The proposed architecture seems not novel, and I have some serious concerns regarding the experiments.


**Summary Of The Paper:**


This paper aims to improve the prediction performance of the protein-protein interaction, which involves predicting partner-specific protein interface contacts (i.e., inter-protein residue-residue contacts) given the 3D tertiary structures of two proteins. In this paper, the authors present a graph transformer to encode the protein geometric features and perform interface contact prediction. The proposed method is validated on two benchmarking datasets, i.e., DIPS-Plus and CASP-CAPRI datasets.


**Summary Of The Review:**

Considering the noticeable improvement made by the authors, I reconsider my original view and agree with the other reviewers on the significance of the proposed approach.

---

> ### Author Response · Authors · 2021-11-19
> **Response to Feedback from Reviewer Qnz5 (1)**
>
> We authors would like to thank you for reading and providing feedback on our manuscript. We have recently uploaded the latest version of our manuscript based on each reviewer's feedback. Above, we have summarized our main changes to the manuscript since its initial submission for review. However, we would like to point out here some specific changes we made based on your review. The following present our most relevant changes concerning your points of feedback.
>
> ---
> Latest Changes to the Manuscript:
>
> 1. We have rewritten Section 4 (Methods) in the manuscript to (1) more clearly highlight the novelties of our new Geometric Transformer compared to the Graph Transformer or other Transformer-like architectures. One way we have done this is by moving much of our discussion of the network modules' design rationales to our manuscript's supplementary material (i.e., Section A - Appendix), to make room in the main text for 5 key new equations describing the operations of the Geometric Transformer's $\textit{Edge}$ $\textit{Initialization}$ $\textit{Module}$ and geometry-evolving $\textit{Conformation}$ $\textit{Module}$. We note that two of these equations describe what we consider to be a primary contribution of our manuscript: the definition and usage of edge geometric neighborhoods in updating geometric representations of 3D graphs. Taken directly from our revised Section 4.2.2 (Conformation Module), the following equations (i.e., Equations 3-5), in our view, succinctly summarize some of the major contributions of our work, demonstrating how we define the edge geometric neighborhood of an edge $e$ to be the $2n$ edges
>
> \\[e_{k} = \\{e_{n_{1}i}, e_{n_{2}j} \mid (n_{1}, n_{2} \\in \\mathbb{N}_{k})\ \\mathrm{and}\ (n_\{1\}, n_\{2\} \\ne i, j)\\}, \\]
>
> where $\mathbb{N}\_{k} \subset \mathbb{N}$ are the source nodes for incoming edges on edge $e's$ source and destination nodes. We then give one of several possible ways in which one can use this definition of an edge geometric neighborhood to update the edge representation $e_{ij}$ of edge $e$:
>
> \\[  O\_{ij}=\sum\_{e\_{ij, k} \in e\_{k}}[f\_{4}(f\_{3}(f\_{2}(f\_{1}(e\_{ij, k}) \odot f\_{\mathrm{f}\_{1}}(\mathrm{f}\_{1, k})) \odot f\_{\mathrm{f}\_{2}}(\mathrm{f}\_{2, k})) \odot f\_{\mathrm{f}\_{3}}(\mathrm{f}\_{3, k})) \odot f\_{\mathrm{f}\_{4}}(\mathrm{f}\_{4, k})]  \\]
>
> \\[  e\_{ij} = \mathrm{RB2}\_{2}(e\_{ij} + \mathrm{RB2}\_{1}(e\_{ij} + O\_{ij})),  \\]
>
> where each term referenced above is defined correspondingly in Section 4.2.2 of our latest manuscript.
>
> To the best of our knowledge, no previous works have considered using the nearest neighborhoods of edges behaving as pseudo-nodes to update the representation of each edge. As such, we have outlined a blueprint from which others can investigate new techniques for using similar edge neighborhoods in future network architectures.
>
> 2. Thank you for pointing out these important citations from Schütt et al. as well as Liu et al. We find these works interesting and certainly related to the problem we have approached in our manuscript. As such, we integrated these citations into our Related Works section (i.e., Section 2 - Page 2, 2nd paragraph) like so: "Previously, geometric learning algorithms like convolutional neural networks (CNNs) and graph neural networks (GNNs) have been used to model molecules and to predict protein interface contacts. Schutt et al. (2017) introduced a deep tensor neural network designed for molecular tasks in quantum chemistry." (Later on in the Related Works section): "Liu et al. (2021) created a graph neural network for predicting the effects of mutations on protein-protein binding affinities, and, more recently, Costa et al. (2021) introduced a Euclidean equivariant transformer for protein docking. Both of these methods may benefit from the availability of precise interface predictors by using them to generate contact maps as input features."
>
> 3. As you suggested, we have now included experiments on the Docking Benchmark 5 dataset to allow others to easily port their future results for comparison as this dataset has traditionally been used to assess the performance of interface contact predictors. Additionally, in the supplementary material (i.e., Section A - Appendix), we include two tables, listing the PDB and chain IDs necessary to reproduce our experiments on the publicly-available DIPS-Plus and CASP-CAPRI datasets, respectively. We have also included the top-k recall metric to complement our experiments' top-k precision scores, to further elucidate the behavior of our models on each protein complex dataset we use for benchmarking.
>
> ---
> We would like to thank you in advance for any additional feedback you may have for our manuscript as it currently stands. Thank you once again for your time and constructive comments!

---

> > ### Comment · Reviewer_Qnz5 · 2021-11-22
> > **The names of the functions need further consideration.**
> >
> > I have read the author's response, as well as the other reviewers' comments. I have noticed that the author add several key equations to describe the proposed method. I am happy to see it since precisely presenting the computations is critical for a methodological paper. One additional issue is that the names of the functions should be further improved.
> >
> > Considering the noticeable improvement made by the authors, I reconsider my original view and agree with the other reviewers on the significance of the proposed approach. I have adjusted my score accordingly.

---

> > > ### Author Response · Authors · 2021-11-27
> > > **Improvement to Function Names**
> > >
> > > Dear Reviewer Qnz5,
> > >
> > > We authors have finished making improvements to our equations and figures describing the Geometric Transformer and its Conformation Module, thanks to your suggestion. The changes we have made are shown below.
> > >
> > > ---
> > > Changes Made to Equations and Figures:
> > >
> > > 1. We have rewritten the equations and architecture descriptions given in Section 4.2.1 (Edge Initialization Module) as follows.
> > >
> > > "To accelerate its training, the Geometric Transformer first embeds each edge $e \in \mathbb{E}$ with the initial edge representation
> > >
> > > \\[
> > > c\_{ij} = \phi\_{e}^{1}([\mathrm{p}\_{1}\ ||\ \mathrm{p}\_{2}\ ||\ \phi\_{e}^{\mathrm{m}\_{ij}}(\mathrm{m}\_{ij}\ ||\ \lambda\_{e})\ ||\ \phi\_{e}^{\mathrm{f}\_{1}}(\mathrm{f\_{1}})\ ||\ \phi\_{e}^{\mathrm{f}\_{2}}(\mathrm{f\_{2}})\ ||\ \phi\_{e}^{\mathrm{f}\_{3}}(\mathrm{f\_{3}})\ ||\ \phi\_{e}^{\mathrm{f}\_{4}}(\mathrm{f\_{4}})])\ \ \ \ \ (1)
> > > \\]
> > >
> > > \\[
> > > e\_{ij} = \phi\_{e}^{2}(\rho\_{e}^{a}(\rho\_{e}^{g}(c\_{ij})))\ \ \ \ \ (2)
> > > \\]
> > >
> > > where $\phi\_{e}^{i}$ refers to the $i$'th edge information update function such as a multi-layer perceptron; $||$ denotes channel-wise concatenation; $\mathrm{p}\_{1}$ and $\mathrm{p}\_{2}$, respectively, are trainable one-hot vectors indexed by $P\_{i}$ and $P\_{j}$, the positions of nodes $i$ and nodes $j$ in the chain's underlying amino acid sequence; $\mathrm{m}\_{ij}$ are any user-predefined features for $e$ (in our case the normalized Euclidean distances between nodes $i$ and nodes $j$); $\lambda\_{e}$ are $\textit{edge-wise}$ sinusoidal positional encodings $sin(P\_{i} - P\_{j})$ for $e$; $\mathrm{f}\_{1}$, $\mathrm{f}\_{2}$, $\mathrm{f}\_{3}$, and $\mathrm{f}\_{4}$, in order, are the four protein-specific geometric features defined in Section A.3; and $\rho\_{e}^{a}$ and $\rho\_{e}^{g}$ are feature addition and channel-wise gating functions, respectively."
> > >
> > > ---
> > > 2. We have rewritten a few introductory portions of Section 4.2.2 (Conformation Module) as follows.
> > >
> > > "Precisely, $\mathbb{E}_{k}$, the edge geometric neighborhood of $e$, is defined as the $2n$ edges
> > >
> > > \\[
> > > \mathbb{E}\_{k} = \\{e\_{n\_{1}i}, e\_{n\_{2}j} \mid (n\_{1}, n\_{2} \in \mathbb{N}\_{k})\ \mathrm{and}\ (n\_{1}, n\_{2} \ne i, j)\\},\ \ \ \ \ (3)
> > > \\]
> > >
> > > where $\mathbb{N}_{k} \subset \mathbb{N}$ are the source nodes for incoming edges on edge $e's$ source and destination nodes."
> > >
> > > ---
> > > 3. We have rewritten a handful of intermediate portions of Section 4.2.2 (Conformation Module) as follows.
> > >
> > > "In the conformation module, the iterative processing of all geometric neighborhood features for edge $e$ can be represented as
> > >
> > > \\[
> > > O\_{ij}=\sum\_{k \in \mathbb{E}\_{k}} \ [(\phi\_{e}^{n}(e\_{ij, k}^{n}) \odot \phi\_{e}^{\mathrm{f}\_{n}}(\mathrm{f}\_{n})), \forall n \in \mathbb{F}]\ \ \ \ \ (4)
> > > \\]
> > >
> > > \\[
> > > e\_{ij} = 2 \times ResBlock\_{2}(\phi\_{e}^{5}(e\_{ij}) + 2 \times ResBlock\_{1}(\phi\_{e}^{5}(e\_{ij}) + O\_{ij})),\ \ \ \ \ (5)
> > > \\]
> > >
> > > where $\mathbb{F}$ are the indices of the geometric features $\\{\mathrm{f}\_{1}, \mathrm{f}\_{2}, \mathrm{f}\_{3}, \mathrm{f}\_{4}\\}$ defined in Section A.3; $\odot$ is element-wise multiplication; $e\_{ij, k}^{n}$ is neighboring edge $e\_{k}$'s representation after gating with $\mathrm{f\_{n - 1}}$; and $2 \times ResBlock\_{i}$ represents the $i$'th application of two unique, successive residual blocks, each defined as $ResBlock(x)=\phi\_{e}^{Res\_{2}}(\phi\_{e}^{Res\_{1}}(x)) + x."$

---

> > > ### Author Response · Authors · 2021-11-29
> > > **Update to Protein Geometric Feature References**
> > >
> > > Dear Reviewer Qnz5,
> > >
> > > We thought you should know that, based on recent feedback we received from Reviewer uZ6F, we have accordingly updated our response to your suggestion to improve our equations' function names (https://openreview.net/forum?id=CS4463zx6Hi&noteId=plBIrAuD5k7).
> > >
> > > Thank you for your time and feedback.
> > >
> > > Best,
> > >
> > > Paper3368 Authors

---

> ### Author Response · Authors · 2021-11-19
> **Response to Feedback from Reviewer Qnz5 (2)**
>
> Continuing from our previous list of responses, we would like to mention how we selected the baseline methods against which to compare all DeepInteract models (including those employing the Geometric Transformer).
>
> ---
> In Section 5.3 of our latest manuscript (Selection of Baselines), we outline the criteria by which we selected each baseline method for comparison: "We considered the reproducibility and accessibility of the method as the most important factor for its inclusion in our benchmarks to encourage adoption of accessible and transparent benchmarks for future works. That is, we only selected baseline methods that are easy to reproduce or instead simple for the general public to use to make predictions." Regarding the references [2] and [3] you suggested, to the best of our knowledge, without making substantial modifications to its models and source code, Method [2] does not predict which specific inter-chain residue pairs interact with one another upon binding of the two chains. Instead, it seems to predict cumulative inter-chain binding affinity scores for each complex as a whole, while also not providing its source code for training new models. Regarding Method [3], it would seem that the authors of this work, to the best of our knowledge, have not made their source code, data, or models publicly available, making this method infeasible to authentically reproduce as a baseline. Since neither of these two methods meets our original selection criteria for each baseline (i.e., must be simple to reproduce or instead simple to make predictions with), we decided to omit them from our latest benchmarks and experiments.

---

### Official Review · Reviewer_HqpY · 2021-11-02

**Correctness:** 4
**Technical Novelty And Significance:** 3
**Empirical Novelty And Significance:** 3
**Recommendation:** 6
**Confidence:** 4

**Main Review:**

Strengths:
1. The strengths of this work stem from the architecture used, consisting of attention mechanisms and invariant representations.
2. This work is uses a novel geometric neighborhood representation which considers edges as pseudo-nodes, with geometric feature updates.
3. The exposition and presentation are clear, and the work makes available the data, models, and code.

Weaknesses:
1. The performance metrics and benchmarks are lacking:
- Specifically, the evaluation metric is precision; whereas recall is missing.
- The results are not compared with standard benchmarks, and require porting different results for comparison.
2. The network trained is relatively small and may not scale well.
3. The work does not predict distograms or a sequence of pose updates.

Missing is a reference to recent related work by Costa et al. from MIT:
End-to-end Euclidean equivariant Transformers for protein docking, NeurIPS Workshop on Learning Meaningful Representations of Life, 2021.

Minor typo: Section 5.2 should read "To identify..

**Summary Of The Paper:**

This paper proposes a state of the art deep learning architecture for predicting mutual contacts between proteins.
Specifically, this work presents a Geometric Transformer for rotation and translation invariant protein interface contact prediction,
given 3D protein structures.



**Summary Of The Review:**

In summary, the proposed deep learning architecture for protein interface contact prediction is somewhat new,
though aspects of this contribution exist in related work. The performance metrics used and baselines may be improved.

---

> ### Author Response · Authors · 2021-11-19
> **Response to Feedback from Reviewer HqpY**
>
> We authors would like to thank you for your careful reading and feedback on our manuscript. We have recently uploaded the latest version of our manuscript based on each reviewer's feedback. Above, we have summarized our main changes to the manuscript since its initial submission for review. However, we would like to point out here some specific changes we made based on your review. The following present our most relevant changes concerning your points of feedback.
>
> ---
> Latest Changes to the Manuscript:
>
> 1. We have included the top-k recall metric to complement our experiments' top-k precision scores, to further elucidate the behavior of our models on different types of protein complex datasets.
>
> 2. We have included experiments on the Docking Benchmark 5 dataset to allow others to easily port their future results for comparison. This dataset has traditionally been used to assess the performance of interface contact predictors. Additionally, in the supplementary material (i.e., Section A - Appendix), we include two tables, listing the PDB and chain IDs necessary to reproduce our experiments on the publicly-available DIPS-Plus and CASP-CAPRI datasets, respectively.
>
> 3. Concerning the size of our trained networks, it was restricted by a few factors. First is the variable size of the input protein complexes. Second is the memory-intensive dot-product attention calculations the Geometric Transformer performs. Finally, the hardware available to us for training was limited to 16 GB of GPU memory. These three factors, when considered together, led us to adjust the network's size to address all three factors simultaneously. With access to hardware with more GPU memory, we would like to consider scaling up our models' complexity where suitable.
>
> 4. In our view, the distogram predictions and pose update sequences you suggested would be a fascinating idea in and of itself to explore within our DeepInteract pipeline. For future works, we would like to consider this as a promising direction to investigate. However, due to time limitations, we focused our current efforts on the prediction of static interface contact probability maps for pairs of input proteins.
>
> 5. Thank you for pointing out this citation from Costa et al. We find this work very interesting and certainly related to the problem we have approached in our manuscript. As such, we integrated this citation into our Related Works section (i.e., Section 2 - Page 2, 2nd paragraph) like so:
> "Liu et al. (2021) created a graph neural network for predicting the effects of mutations on protein-protein binding affinities, and, more recently, Costa et al. (2021) introduced a Euclidean equivariant transformer for protein docking. Both of these methods may benefit from the availability of precise interface predictors by using them to generate contact maps as input features."
>
> 6. Thank you for pointing out our small typo in Section 5.2. We have since corrected it to read "To identify" as originally intended.
>
> ---
> We would like to thank you in advance for any additional feedback you may have for our manuscript as it currently stands. Thank you once again for your time and constructive comments!

---

### Official Review · Reviewer_uZ6F · 2021-11-22

**Correctness:** 3
**Technical Novelty And Significance:** 3
**Empirical Novelty And Significance:** 3
**Recommendation:** 6
**Confidence:** 4

**Main Review:**

Strength :
- The approach is original and performs better than a naive GNN implementation.
- Using a more comprehensive data set to train the methods should lead to increased results, it would be an interesting avenue in general to retrain former data-driven methods over this larger data set
- The tackled problem is interesting and the results presented here are claimed to be sota.

Weaknesses :
- Even though the approach is novel, there is not enough evidence that it is needed. Several choices seem to be counter-intuitive and make the approach hard to follow, even with explicit equations. For instance, the attention mechanism is split into successive attention computations. The authors build several geometrical invariant descriptors, why cannot they be simply concatenated into one vector and compute a single attention over this pairwise geometric vector ? In this formulation, I think the approach would more or less reduce to graph transformers.
- The claim that this method could be used on  "other 3D graphs" is too strong because to my understanding, the invariant encoding of the geometric information heavily depends on the ordering of the points. A canonical ordering is not available in general.
- A general equivariant graph based already exists."SE(3)-Transformers: 3D Roto-Translation Equivariant Attention Networks" Fuchs et al. Another more recent and simpler equivariant method can be found in "E(n) Equivariant Graph Neural Networks" (Sattoras et al.). These methods seem to be applicable in your setting here.

- The data used looks very relevant, but since the learning leverages geometric features, I would suggest conducting the splitting of your dataset based on geometric similarity as was done for instance in Maasif (you could use CATH classes or TM-score splitting).
- The choice of the methods against which the tools is benchmarked should be expanded and these methods properly introduced. I think it should include PINET ("Protein interaction interface region prediction by geometric deep learning" Dai et al.) that is to my knowledge the state of the art for this task. Despite what was stated, to my understanding PINET does target-specific prediction. The fact that the performance of the concurrent methods might be corrupted because of improper splitting should be addressed by authors, for instance by removing from the test sets interfaces too close to the training set (see structural splits above).

**Summary Of The Paper:**

This paper addresses contact prediction for PPI by proposing a geometric deep learning framework. They use a k nearest neighbor representation for each protein and compute geometric-based attention scores to convolute the messages over this graph. They validate this approach on the PPI task, using the DIPS-plus dataset.

**Summary Of The Review:**

This paper pursues two complementary aims. A first one is building a predictor for contact residues using the structures of both proteins interacting. To do so, the authors propose leveraging a larger data set along with a novel architecture. However the choice of the methods included in the benchmark is not motivated enough and should include more recent state of the art.

The second one introduces a novel invariant geometry-based attention for use in graph transformers. It also includes using edges as pseudo-nodes in this graph. These ideas seems very interesting and could be validated onto other classical tasks involving protein structure such as binding site prediction. However the chosen formulation sounds a bit complicated and could be motivated by comparing it to a simpler version. Such an analysis should be complemented by a more extensive comparison to existing equivariant methods. I think such a contribution could be a standalone contribution, but the current version of the paper is not motivating this approach enough.

I would advise to reject the paper in the current form, and encourage authors to pursue the validation of their paper along these two avenues.


########################

After an interesting discussion with the authors and inclusion of further results, I changed my evaluations twice and now recommend this paper for acceptance. I would advise the authors to work more on the writing of the paper using the supplementary for the equations and emphasizing more the novelties of their methods compared to previous work (backed up by ablation studies disproving the usefulness of these technical choices).

---

> ### Author Response · Authors · 2021-11-22
> **Response to Feedback from Reviewer uZ6F (1)**
>
> We authors would like to thank you for your feedback on our manuscript. We have recently uploaded the latest version of our manuscript based on your feedback. The following present the most relevant changes we made concerning your points of feedback.
>
> ---
> Latest Changes to the Manuscript in Response:
>
> 1. At the end of Section 2 (Related Work), we have now softened our second proposed contribution of this work (i.e., the second bullet point) to now begin as follows. "We propose the new $\textit{Geometric\ Transformer}$ which can be used for tasks on 3D protein structures and similar biomolecules."
>
> 2. We now include in our manuscript's supplementary material (Section A.5) a discussion of when one may want to use an invariant network architecture for molecular tasks compared to using an equivariant network, to give context as to why we chose to focus on an achieving an invariant representation of protein structures for this task.
>
> 3. We have added in Section 5.3 descriptions of the machine learning algorithms employed by each baseline method.
>
> ---
> Discussion of Concerns Raised:
>
> $\textit{Weakness}$ 1: We would like to thank the reviewers for asking this question. To meet our page limits, we believe much of these concerns raised about the rationale of the Geometric Transformer's design are addressed in our supplementary material (Sections A.6-8). However, we believe it is worth pointing out two specific rationales behind the design of the Geometric Transformer's $\textit{Conformation\ Module}$.
>
> $\textbf{1}$) The first is to have it explicitly incorporate (separate) residual connections back to each geometric feature's original values. Such residual connections then enable what we refer to as the evolution of individual edge geometric features. One potential downside, as we see it, of immediately concatenating each geometric feature into a single tensor is that the network would lose its capacity to perform channel-wise dropout on each original geometric feature value via gating, which may ultimately impact the network's expressiveness by (inadvertently) encouraging it to become overreliant on a subset of features while ignoring others.
>
> ($\textbf{2}$) Second is that, by evolving edge geometric features using neighboring edges, the network is then enabled to perform a form of geometric message passing to update each edge representation. In turn, these updated edge representations will subsequently be injected to update our graphs' node representations (i.e., residue representations) via the Geometric Transformer's dot-product self-attention mechanism. The desired outcome of such operations by the Conformation Module is to further improve each protein graph's node representations, such that interacting inter-protein residue pairs are more clearly distinguished when concatenating their learned representation tensors via channel-wise interleaving.
>
> We believe it is also worth pointing out that the Geometric Transformer includes positional encodings for each edge in a given protein graph, in contrast to the Graph Transformer of Dwivedi and Bresson (2021) which employs only node-wise positional encodings. As such, our work, in effect, explores the use of such edge-wise positional encodings to improve the representations learned by graph-based transformers.
>
> $\textit{Weakness}$ 2: We agree with your assessment. We have since softened this proposed contribution of ours as mentioned in our list of recent revisions above.
>
> $\textit{Weakness}$ 3: We think you have raised a good point for discussion. As such, we have included a paragraph dedicated to this topic in our supplementary material (Section A.5). In addition to this paragraph in our supplementary material, we would like to note that, based on our previous experiments with the SE(3)-Transformer and similar attention-based equivariant networks, such networks, when applied to large biomolecules such as protein complexes, can prohibitively exceed the memory capacity of some of the most commonly used GPUs for training deep learning models (e.g., 16GB Nvidia Tesla V100 GPUs such as the ones available to us for training), thereby significantly affecting the size of protein complexes one can include in one's training dataset.
>
> $\textit{Weakness}$ 4: We thank you for this idea of geometric similarity splitting. Based on our understanding, such geometry-based splitting is a fairly recent idea and, as such, has not yet been standardized in the bioinformatics community (e.g., Drug-Target Affinity Prediction - Jiang et al. (2020); Quality Assessment of Protein Docking Models - Han et al. (2021)). We would still like to introduce additional filtering methods such as this into our DeepInteract pipeline in the near future. However, due to the resource and time constraints imposed by our forthcoming rebuttal deadline (i.e., today), we must defer an exploration of such ideas to future work.
>
> ---
> Thank you in advance for any additional feedback you may have for our manuscript.

---

> > ### Comment · Reviewer_uZ6F · 2021-11-23
> > **Continued review**
> >
> > I thank the authors for their fast, detailed and accurate answers.
> >
> > Let me first comment on the methodological contribution :
> > 1. First of all, the appendices A.5-8 are not referenced in the main text, which makes the intuitive motivation of the method a bit hard to access.
> >
> > For Your 1.1, I do not understand the part about residual connections that also happen in GraphTransformer, but I do see that the gating procedure is different.
> > Your 1.2 is unclear to me, blending node and edge representation was extensively used in AlphaFold2 or GraphTransformers to my understanding.
> > The way I understand it, the major improvements conducted over GraphTransformers are A. inclusion of edge positional encoding, B. usage of pseudo-nodes, C. inclusion of invariant geometric features, D. a specific gating mechanism.
> > I think points A and B are easy enough to explain. C. is also interesting (see my next point) and an ablation study was conducted to disprove its usefulness. D. however motivates most of the complexity of the paper and does not convince me. It motivates most of the figures and generates unintuitive equations (Fig 3., 4. and equations 1., 2., 4. and 5.) to understand the intricate combinations used. Moreover, that gating induce an arbitrary ordering of the geometric features to my understanding. This gating mechanism is motivated by a paper, that despite not having read in great details, does not seem to be directly applied to our setting here. Moreover its focus is on controlling the vanishing gradients in RNNs, while you apply this gating mechanism only once per layer, and you have two layers. My intuition may be wrong here, but the usefulness of this complexity generating subtlety needs more motivation, for instance with an ablation study. If you implement a simple concatenation of the features with C, are the results affected ?
> >
> > 2 and 3. I appreciate the note A.5 and propose a little reflection about the role of equivariance and invariance here.
> > First of all, the memory concerns are a real limitation to use equivariant transformers methods, however, E(n) Equivariant Graph Neural Networks, should not suffer as much from it.
> >
> > Using equivariance throughout the network has repeatedly been shown to yield better results than enforcing a direct invariance of the learnt features, encoding the relative pose of features (see for instance "Steerable CNNs"). For this reason, I object that your A.5 is not very adapted to your paper.
> >
> > Actually the reason you obtain good expressivity relies heavily onto the ordering of the points which was my objection 2. This property of proteins enable choosing a canonical set of local frames, efficiently encoding these relative poses. This could side step the need for equivariant methods altogether and, along with the memory argument, consist a reasonable way to not try other equivariant methods. However, this idea of using local frames leveraging the ordering of the points is not novel as it was, for instance, used in AlphaFold2.
> > Overall, I thus think the part about invariance should thus be rewritten to underline how for protein or ordered sets, this trick can yield invariant networks in a cheap and efficient way.
> >
> >
> > Taking a step back, I think the main contributions are the inclusion of geometric features in a transformer graph model and the application of this modified model to a specific problem. The other methodological contributions (gating, positional encoding) are not well-grounded and empirically-validated through ablation.
> >
> > I will post another comment about the application later.

---

> > > ### Author Response · Authors · 2021-11-24
> > > **Response to Continued Review (Part 1)**
> > >
> > > We authors are grateful for your latest feedback on and discussion of our paper. We have briefly made some of your recommended revisions to our local copy of the paper, as detailed below (with more to follow soon).
> > >
> > > ---
> > > Changes Made (or Currently Being Made) to Our (Local) Manuscript:
> > >
> > > 1. As follows, to more directly convey the rationale behind the design of the Geometric Transformer, we now explicitly refer readers in Section 4.2 (Geometric Transformer Architecture) of our main text to our supplementary material sections A.6-8 for an expanded discussion on the model's design:
> > >
> > > "Hypothesizing that a self-attention mechanism that evolves proteins' physical geometries is a key component missing from existing interface contact predictors, we propose the Geometric Transformer, a graph neural network explicitly designed for capturing and iteratively \textit{evolving} protein geometric features. As shown in Figure 3, the Geometric Transformer expands upon the existing Graph Transformer architecture (Dwivedi et al. (2021)) by introducing ($\textbf{1}$) an edge initialization module, ($\textbf{2}$) positional encoding for edges, and ($\textbf{3}$) a geometry-evolving conformation module (see Sections A.6, A.7, and A.8 for rationale)."
> > >
> > > 2. We agree that two ablation studies concerning our edge positional encodings and geometric feature gating would be beneficial in elucidating the contribution of each network component in the Geometric Transformer. As such, we are currently performing these experiments and will post them as a comment here on OpenReview once these results are ready.
> > >
> > > 3. Concerning the discussion of equivariance compared invariance in our network architecture, we agree that perhaps a better way of presenting the Geometric Transformer's use of invariance to enhance network expressiveness is to instead describe our formulation of network invariance as a convenient and efficient alternative to traditionally-expensive equivariant attention networks made possible by the canonical ordering of points present in data domains such as ordered sets and proteins. As such, we have revised the last few sentences of Section 4.2.2 (Conformation Module) to read as follows:
> > >
> > > "Described in Section A.3, by way of their construction, each of our selected edge geometric features is translation and rotation invariant to the network's input space. As discussed in Section A.5, we couple these features with our choice of node-wise positional encodings (see Section 4.2.3) to attain canonical invariant local frames for each residue to encode the relative poses of features in our protein graphs. In doing so, we leverage many of the benefits of employing equivariant representations while reducing the large memory requirements they typically induce, to yield a robust invariant representation of each input protein."
> > >
> > > In our revised Section A.5 (Equivariance or Invariance?), we then rationalize our choice of invariance as follows:
> > >
> > > "In our view, a natural question to ask concerning a deep learning architecture designed for a specific task is whether equivariance to translations and rotations in $\mathbb{R}^{3}$ should be preferred over invariance to transformations in such a geometric space. The benefits of employing equivariant representations in a deep learning architecture primarily include symmetry-preserving updates to type-1 tensors such as the coordinates representing an object in $\mathbb{R}^{3}$ and the derivation of invariant relative feature poses for type-0 features such as scalars (Cohen et al. (2016)). However, equivariant representations, particularly those derived with a self-attention mechanism, typically induce large memory requirements for training and inference. In contrast, in the context of data domains such as ordered sets or proteins where there exists a canonical ordering of points, invariant representations may be adopted to simultaneously reduce memory requirements and provide many of the benefits of using equivariant representations such as attaining these relative poses of type-0 features (Ingraham et al. (2019) and Jumper et al. (2021)). As such, in the context of the Geometric Transformer, we decided to pursue invariance over equivariance, to reduce the network's effective memory requirements and to improve its learning efficiency and generalization capabilities (Bronstein et al. (2021)). However, for applications such as protein-protein docking that may more directly rely on type-1 tensors for network predictions (Costa et al. (2021)), designing one's network architecture to preserve full translation and rotation equivariance in $\mathbb{R}^{3}$ is, in our perspective, a worthwhile research direction to pursue as many promising results on molecular datasets have already been demonstrated with equivariant neural networks such as SE(3)-Transformers (Fuchs et al. (202)) and lightweight graph architectures such as the Equivariant Graph Neural Network (Satorras et al. (2021))."

---

> > > ### Author Response · Authors · 2021-11-25
> > > **Ablation Study concerning Edge Positional Encoding (EPE)**
> > >
> > > We authors have recently collected results for our ablation study of the sinusoidal edge positional encodings in the Geometric Transformer, to see what effect removing them has on the Geometric Transformer's top-$k$ precision and recall for all three datasets we consider in our paper (i.e., DIPS-Plus, CASP-CAPRI, and DB5). Below, we have added one new row to each of our seven results tables to indicate the results for this ablation study (i.e., Geometric Transformer without edge positional encoding = GeoT w/o EPE).
> > >
> > >
> > > $Table\ 1$ (Section 5.3 - New DIPS-Plus Experiment):\
> > > DI (GeoT w/o EPE):         $\mathbf{0.28}$ ($\mathbf{0.05}$)       &   0.24 (0.01)     & 0.23 (0.03)    &  0.11 (0.05)       &    0.10 (0.04)      & 0.09 (0.03)
> > >
> > > $Table\ 2$ (Section 5.3 - New DIPS-Plus Experiment):\
> > > DI (GeoT w/o EPE):         0.19 (0.04)       &   0.18 (0.03)     & 0.16 (0.03)    &  0.14 (0.02)       &    0.08 (0.02)      & 0.04 (0.02)
> > >
> > > $Table\ 3$ (Section 5.3 - New CASP-CAPRI Experiment):\
> > > DI (GeoT w/o EPE):         0.11 (0.01)    &     0.12 (0.02)     &     0.11 (0.01)   &   0.18 (0.07)     &      0.20 (0.09)    &      0.18 (0.04)
> > >
> > > $Table\ 4$ (Section 5.3 - New CASP-CAPRI Experiment):\
> > > DI (GeoT w/o EPE):         0.13 (0.02)      &   0.14 (0.03)      &    0.13 (0.02)    &  0.12 (0.01)       &    0.07 (0.01)      &    0.03 (0.01)
> > >
> > > $Table\ 5$ (Section 5.3 - New DB5 Experiment):\
> > > DI (GeoT w/o EPE):     0.011 (0.004)       &    0.009 (0.004)      &    0.011 (0.002)    &   0.018 (0.01)   &   0.010 (0.004)    &    0.0034 (0.002)
> > >
> > > $Table\ 6$ (Section A.2 - New DIPS-Plus Experiment):\
> > > DI (GeoT w/o EPE):         0.18 (0.02)       &   0.11 (0.01)     & 0.05 (0.01)    &  0.11 (0.03)       &    0.07 (0.02)      & 0.03 (0.02)
> > >
> > > $Table\ 7$ (Section A.2 - New CASP-CAPRI Experiment):\
> > > DI (GeoT w/o EPE):         0.11 (0.01)       &   0.07 (0.01)     & 0.04 (0.01)    &  0.12 (0.02)       &    0.07 (0.01)      & 0.03 (0.01)
> > >
> > > In summary, through these seven experiments, we see that the addition of a sinusoidal edge positional encoding $\textit{noticeably}$ improves both top-$k$ precision and recall across our three selected datasets. However, in many cases, there remains a performance gap between the standard Geometric Transformer (i.e., one with both geometric feature gating and sinusoidal edge positional encodings) and our EPE-ablated Geometric Transformer, suggesting that the addition of edge positional encodings is not the only architectural design choice responsible for the standard Geometric Transformer's high performance across all three datasets.
> > >
> > > We are currently conducting our ablation study of the geometric feature gating (GFG) within the Geometric Transformer (i.e., instead directly concatenating geometric features). We will soon post those results here in a similar manner.

---

> > > ### Author Response · Authors · 2021-11-27
> > > **Ablation Study concerning Geometric Feature Gating (GFG)**
> > >
> > > We authors have recently collected results for our ablation study of the geometric feature gating scheme the Geometric Transformer employs, to see what effect removing it has on the Geometric Transformer's top-$k$ precision and recall for all three datasets we consider in our paper (i.e., DIPS-Plus, CASP-CAPRI, and DB5). Below, we have added one new row to each of our seven results tables to indicate the results for this ablation study (i.e., Geometric Transformer without geometric feature gating = GeoT w/o GFG).
> > >
> > >
> > > $Table\ 1$ (Section 5.3 - New DIPS-Plus Experiment):\
> > > DI (GeoT w/o GFG):         0.27 (0.08)       &   0.24 (0.08)     & 0.21 (0.08)    &  0.10 (0.02)       &    0.09 (0.02)      & 0.09 (0.01)
> > >
> > > $Table\ 2$ (Section 5.3 - New DIPS-Plus Experiment):\
> > > DI (GeoT w/o GFG):         0.18 (0.05)       &   0.16 (0.04)     & 0.15 (0.04)    &  0.14 (0.02)       &    0.08 (0.02)      & 0.04 (0.01)
> > >
> > > $Table\ 3$ (Section 5.3 - New CASP-CAPRI Experiment):\
> > > DI (GeoT w/o GFG):         0.10 (0.02)    &     0.10 (0.02)     &     0.09 (0.02)   &   0.14 (0.03)     &      0.17 (0.03)    &      0.14 (0.02)
> > >
> > > $Table\ 4$ (Section 5.3 - New CASP-CAPRI Experiment):\
> > > DI (GeoT w/o GFG):         0.11 (0.01)      &   0.12 (0.02)      &    0.10 (0.02)    &  0.11 (0.01)       &    0.06 (0.01)      &    0.03 (0.01)
> > >
> > > $Table\ 5$ (Section 5.3 - New DB5 Experiment):\
> > > DI (GeoT w/o GFG):     0.008 (0.001)       &    0.008 (0.001)      &    0.009 (0.002)    &   0.014 (0.01)   &   0.006 (0.002)    &    0.003 (0.001)
> > >
> > > $Table\ 6$ (Section A.2 - New DIPS-Plus Experiment):\
> > > DI (GeoT w/o GFG):         0.19 (0.04)       &   0.11 (0.03)     & 0.05 (0.02)    &  0.09 (0.01)       &    0.05 (0.02)      & 0.03 (0.01)
> > >
> > > $Table\ 7$ (Section A.2 - New CASP-CAPRI Experiment):\
> > > DI (GeoT w/o GFG):         0.10 (0.02)       &   0.06 (0.01)     & 0.03 (0.01)    &  0.11 (0.02)       &    0.07 (0.01)      & 0.03 (0.01)
> > >
> > > In summary, through these seven experiments, we see that the addition of geometric feature gating within the Geometric Transformer $\textit{considerably}$ improves both top-$k$ precision and recall across our three selected datasets by evolving protein geometric features with each Geometric Transformer layer. For instance, we observe a sizeable performance gap between the standard Geometric Transformer (i.e., one with both geometric feature gating and sinusoidal edge positional encodings) and our GFG-ablated Geometric Transformer, suggesting that the addition of geometric feature gating is one of the key architectural design choices responsible for the standard Geometric Transformer's high performance across all three datasets. We believe such results, coupled with those of our previous ablation study on edge positional encodings, support the conclusion that both $(\mathbf{1})$ a sinusoidal edge positional encoding and $(\mathbf{2})$ geometric feature gating contribute noticeable gains in precision and recall across a variety of protein complex types.

---

> > > > ### Comment · Reviewer_uZ6F · 2021-11-29
> > > > **Good job !**
> > > >
> > > > Hello,
> > > >
> > > > You have made your point. I am still surprised by the performance gap induced by inclusion of the gating procedure which I had not read about a lot. However this empirical evidence convinces me. I salute your efforts to make the appropriate modifications and change my evaluation.
> > > >
> > > > As a minor suggestion, I think you should not rename f4-7 to f1-4 in the paper as it is confusing and not grounded in my point of view.
> > > >
> > > > I think our discussion and these ablation studies improved your paper, and hope that you feel the same way.
> > > >
> > > > Best regards

---

> > > > > ### Author Response · Authors · 2021-11-29
> > > > > **Update Based on Feedback**
> > > > >
> > > > > Dear Reviewer uZ6F,
> > > > >
> > > > > Based on your helpful suggestion, we have updated our local copy of the manuscript such that Figure 3 (Geometric Transformer Overview) and its accompanying equations (Equations 1 and 2) read as they appear in our recent comment to Reviewer Qnz5 (https://openreview.net/forum?id=CS4463zx6Hi&noteId=plBIrAuD5k7). In short, we believe we have reduced the chance of confusing readers when we reference our protein geometric features by reindexing them to start from 1, for our description of the Edge Initialization Module as well as our description of the Conformation Module.
> > > > >
> > > > > Thank you again for your helpful feedback.
> > > > >
> > > > > Best,
> > > > >
> > > > > Paper3368 Authors

---

> > ### Comment · Reviewer_uZ6F · 2021-11-23
> > **Continued review (part 2)**
> >
> > Let us now talk about the application side.
> >
> > 4. I agree that this splitting is not the most common one, though it is also discussed in a work you cite : MaSIF (Gainza et al.). I also think the concern was the converse of my objection about splitting : you benchmark against methods trained on other databases, which means that these might have a much lower structural similarity to the test set than the one you used, making it an unfair comparison.
> >
> > 5. Thanks for introducing the methods. Actually, I think the whole paragraph you wrote "We note that we also considered adding more recent baseline methods such as those of Townshend et al. (2019) and Liu et al. (2020). However, for both of these methods, we were not able to locate any provided source code or web server predictors facilitating the prediction of inter-protein residue-residue contacts for provided FASTA or PDB targets, so they ultimately did not meet our baseline selection criterion of reproducibility/an ability to make predictions." should be included in the paper. This is the complete justification of what methods are to be benchmarked and incentivizes methods to be reproducible.
> > You are right about the inclusion of PINET, it is indeed partner specific but does not offer contact prediction. Therefore, the results appear more solid to me. I am a bit surprised about the very poor precision performance of BIPSI on CASP that do not resemble the ones reported
> > by the authors of DeepHomo.
> >
> > I think overall, the results paragraph makes sense but is not detailed enough and this application contribution does not belong in ICLR per se. I have changed my gradings to reflect that these results appear to be good.
> >
> > I once again thank the authors for their work and answers.

---

> > > ### Author Response · Authors · 2021-11-24
> > > **Response to Continued Review (Part 2)**
> > >
> > > We now turn to the applications side of our work.
> > >
> > > ---
> > > 1. Based on what we authors have observed in this particular subfield of bioinformatics, standardization of benchmarking data, methods, and metrics could be improved considerably. Towards this direction, part of our work's contributions is that we introduce standardized ranking metrics such as top-$k$ precision and recall in our benchmark results that we believe (based on our experience, for example, in intra-chain protein contact prediction) more clearly illustrate the real-world performance of each model included in our results section, especially when they are used to select inter-chain residue pairs that are most likely to be in contact with one another $\textit{after\ two\ chains\ bind\ together}$. We agree that including new data splitting techniques would be a step towards standardization in this particular subfield, and we are committed to working towards this goal. For example, we are interested in making open-source contributions to datasets such as DIPS-Plus to include in them some of these new data splitting techniques discussed here on OpenReview. We believe making such contributions could encourage others to use a central database like DIPS-Plus in the training of future interface contact prediction models to be developed, ultimately to facilitate a more convenient comparison of contact prediction models. Of course, such efforts will require dedicated time and resources. As previously discussed, given the time limits imposed on us by our rebuttal deadline, we are not able to incorporate such efforts at this moment and will factor them in for future works.
> > >
> > > 2. In addition, we would like to note that our results on the Docking Benchmark 5 (DB5) dataset may be able to address some of your concerns regarding the geometric similarity between proteins in our training dataset as each protein complex in DB5 is known to be well-diversified with respect to SCOP families (Vreven et al. (2015)). Further, we made great efforts to ensure there is minimal sequence-based overlap between our training dataset, DIPS-Plus, and DB5 to ensure our results are comparable with those of other methods.
> > >
> > > ---
> > > Concerning our recent revisions, below we describe the revisions we have made on the applications side of our manuscript based on your feedback.
> > >
> > > ---
> > > Change(s) Made to Our (Local) Manuscript:
> > >
> > > 1. Based on your excellent feedback, we have revised our first paragraph in Section 5.3 (Selection of Baselines) to read as follows:
> > >
> > > "We considered the reproducibility and accessibility of a method to be the most important factors for its inclusion in our following benchmarks to encourage the adoption of accessible and transparent benchmarks for future works. As such, we have included the methods BIPSPI (an XGBoost-based algorithm) (Sanchez-Garcia et al. (2018)), DeepHomo (a CNN for homodimers) (Yan et al. (2021)), and ComplexContact (a CNN for heterodimers) (Zeng et al. (2018)) since they are either easy to reproduce or simple for the general public to use to make predictions. Each method predicts interfacing residue pairs subject to the (on average) $1$:$1000$ positive-negative class imbalance imposed by the biological sparsity of true interface contacts. We note that we also considered adding more recent baseline methods such as those of Townshend et al. (2019) and Liu et al. (2020). However, for both of these methods, we were not able to locate any provided source code or web server predictors facilitating the prediction of inter-protein residue-residue contacts for provided FASTA or PDB targets, so they ultimately did not meet our baseline selection criterion of reproducibility (e.g., an ability to make new predictions)."
> > >
> > > ---
> > > Lastly, we would like to offer our perspective on why our current paper distinctly fits into the core relevant topics of ICLR. Firstly, to the best of our knowledge, on ICLR's website (https://iclr.cc/) applications in computational biology are explicitly listed and welcomed for review. Namely, the website states the following:
> > >
> > > "A non-exhaustive list of relevant topics explored at the conference include:
> > >
> > > ...
> > > - applications in audio, speech, robotics, neuroscience, $\textit{computational biology}$, or any other field"
> > >
> > > Additionally, over the last couple of years, we have seen exemplary examples of pioneering research at the intersection of deep learning and bioinformatics published here at ICLR. For example, ICLR papers such as "Intrinsic-Extrinsic Convolution and Pooling for Learning on 3D Protein Structures" by Hermosilla et al. (2021) have demonstrated significant advances in deep learning for proteins, specifically. We believe such evidence validates our choice of selecting ICLR for submission of our work.
> > >
> > > ---
> > > Once again, we would like to thank you for your insightful feedback on our paper.

---

> ### Author Response · Authors · 2021-11-22
> **Response to Feedback from Reviewer uZ6F (2)**
>
> Continuing from our previous list of responses:
>
> ---
> Discussion of Concerns Raised:
>
> $\textit{Weakness}$ 5.1: Thank you for this suggestion about PINET. We have since reassessed this work and its accompanying code repository. However, after reviewing all available source code and documentation, we are not able to identify any capabilities of the PINET model to predict specifically which inter-protein $\textit{residue\ pairs}$ are in contact with one another upon binding. Instead, it seems as though the model can predict which residues in both proteins are likely to fall within the interaction interface region. Hence, this method addresses a slightly different problem than the one DeepInteract is designed to solve. For example, our formulation of interface contact prediction imposes nearly a 1:1000 positive-negative class imbalance on deep learning models, whereas PINET faces a much smaller class imbalance of 8%, since interface contact prediction is, by design, a more specific task than interaction interface region prediction.
>
> To your point about baseline methods needing to be properly introduced, we have noted this and added to Section 5.3 (Selection of Baselines) descriptions of the types of machine learning algorithms employed by each baseline method included in our benchmarks (i.e., BIPSPI, DeepHomo, and ComplexContact). We note that we also considered adding more recent baseline methods such as those of Townshend et al. (2019) and Liu et al. (2020). However, for both of these methods, we were not able to locate any provided source code or web server predictors facilitating the prediction of inter-protein residue-residue contacts for provided FASTA or PDB targets, so they ultimately did not meet our baseline selection criterion of reproducibility/an ability to make predictions.
>
> $\textit{Weakness}$ 5.2: We would like to note that this (possible) occurrence of data leakage concerning ComplexContact's training dataset is in regards to the DIPS-Plus dataset since it is derived directly from the RCSB PDB. We primarily intend for the results shown for the DIPS-Plus dataset to complement the results shown on the (arguably more important) CASP-CAPRI 13-14 and DB5 datasets. We included them to further demonstrate the Geometric Transformer's ability to balance achieving state-of-the-art (or competitive) results for both types of protein complexes, not for readers to consider the DIPS-Plus dataset metrics in isolation from other dataset results. In this spirit, our results on the CASP-CAPRI and DB5 datasets are intended to add more generality to our results set since the data splitting criteria we employ follow standard practices in the bioinformatics community such that our results more closely reflect application scenarios of DeepInteract. We agree with your related point in Weakness 4 that adding novel data splitting strategies such as the one you suggested would be interesting to consider, notably for ablation studies. However, due to the rebuttal time constraints imposed on us authors, we are not able to include such studies in this work but would like to include them in future works.
>
> ---
> Included below, for reviewers' convenience, is our new discussion of invariance vs. equivariance in Supplementary Section A.5 (to complement our rebuttal concerning the reviewer's third provided weakness).
>
> "In our view, a natural question to ask concerning a deep learning architecture designed for a specific task is whether invariance to translations and rotations in $\mathbb{R}^{3}$ should be preferred over full equivariance to transformations in such a geometric space. Our answer to this question, in the context of the Geometric Transformer, is that for tasks such as static interface prediction between proteins, invariance should be preferred over equivariance as the benefits of equivariance primarily include equivariant updates to type-1 tensors such as the coordinates representing an object in $\mathbb{R}^{3}$. Our formulation of the interface prediction problem solely makes use of the features derived for each residue in our protein graphs to calculate our models' cross entropy loss. Therefore, in this context, we instead decided to focus on preserving translation and rotation invariance in $\mathbb{R}^{3}$ to improve the learning efficiency and generalization capabilities of the Geometric Transformer (Bronstein et al. (2021)). However, for applications such as protein-protein docking that may more directly employ type-1 tensors for network predictions (Costa et al. (2021)), designing one's network architecture to preserve full equivariance in $\mathbb{R}^{3}$ is, in our perspective, a worthwhile research direction to pursue as many promising results on molecular datasets have already been demonstrated with equivariant neural networks such as SE(3)-Transformers (Fuchs et al. (2020)) and Equivariant Graph Neural Networks (Satorras et al. (2021))."

---

### Author Response · Authors · 2021-11-19
**Summarized Responses to Reviewers' Feedback (i.e., Reviewers HqpY, Qnz5, and rPiF)**

We authors would like to thank all the reviewers for carefully reading through and offering insightful feedback to us concerning our manuscript. We believe this feedback has significantly improved the quality of the manuscript since its submission for initial review.

For simplicity and expediency, below we have highlighted the major points of feedback we have aimed to address in our latest version of this manuscript (as of now uploaded here on OpenReview). As always, we welcome any additional feedback on the changes we have made in our latest revisions.

---
Latest Changes to the Manuscript:

1. We have added a new set of experiments on Docking Benchmark 5 (DB5), a widely-used and challenging dataset for interface contact predictors. Such results demonstrate that the Geometric Transformer enhances the ability of DeepInteract, our interface contact prediction pipeline, to predict inter-chain residue-residue interactions for geometrically-complex heteromeric proteins in their $\textit{conformationally-unbound}$ state, which is an end goal for interface contact predictors.

2. All of our experiments, including those on DB5, now report both the top-k precision and the top-k $recall$ metrics frequently found in combination with each other (e.g., when assessing the ranking performance of recommender systems).

3. We have rewritten Section 4 (Methods) in the manuscript to (1) more clearly highlight the novelties of our new Geometric Transformer compared to the Graph Transformer or other Transformer-like architectures. One way we have done this is by moving much of our discussion of the network modules' design rationales to our manuscript's supplementary material (i.e., Section A - Appendix), to make room in the main text for 5 key new equations describing the operations of the Geometric Transformer's $\textit{Edge}$ $\textit{Initialization}$ $\textit{Module}$ and geometry-evolving $\textit{Conformation}$ $\textit{Module}$. We note that two of these equations describe what we consider to be a primary contribution of our manuscript: the definition and usage of edge geometric neighborhoods in updating geometric representations of 3D graphs.

4. We updated the figures in the main text to simplify and (ideally) make them easier to read and understand.

5. We have re-run all experiments and recalculated their results after correcting a few small bugs affecting our DeepInteract models' evaluation scripts (i.e., affecting their top-k precision scores). We note that our latest set of results seems to suggest that the Geometric Transformer adds considerable improvements to top-k precision on heteromeric proteins (i.e., protein complexes with asymmetric inter-chain geometries) from the CASP-CAPRI and DB5 datasets while maintaining state-of-the-art or competitive results on homomeric proteins from all datasets considered.

6. We do note that, in our experiments on the DIPS-Plus dataset, we observe the method ComplexContact (CC) outperform the performance of all of our DeepInteract models, including those employing the Geometric Transformer. We believe this may be explained by the fact that ComplexContact's deep learning model has likely been trained on a least a few of the DIPS-Plus proteins designated for testing (or those with high sequence similarity to such proteins), since these protein targets are publically available via the RCSB and, as such, may be present in similar protein complex datasets for machine learning. Similarly, we note that the Graph Transformer of Dwivedi and Bresson (2021) and the Graph Convolutional Network of Kipf et al. (2016), for a single top-k precision category each, marginally exceed the performance of the Geometric Transformer when used within DeepInteract on the DIPS-Plus and CASP-CAPRI datasets, respectively. These two instances only occur on homodimer protein targets, so we believe a reasonable explanation for this may be that there exists a trade-off in some of these deep learning models between optimizing precision for homomeric proteins and optimizing precision for heteromeric proteins. If such a hypothesis is true, it would seem that the Geometric Transformer has prioritized its precision on heteromeric protein targets slightly more than its precision on homomeric targets. In our experience, in a real-world setting, many bioinformaticians and other life science researchers would prefer to see their models optimizing their precision on heteromeric proteins since these targets are often considered to be much harder targets for deep learning algorithms to correctly model.

7. We have added (Section A.5) a discussion of the use of invariance and equivariance for different tasks.

8. We have softened our previous claim on the general applicability of the Geometric Transformer for 3D graphs.

9. We have added a description of the machine learning algorithm(s) employed in each of our baseline methods.

---
We would like to thank all the reviewers for their time and constructive comments!

---

### Comment · Area_Chair_ww9s · 2021-11-22
**A fourth review is in**

Dear authors and reviewers,
Since this paper only had 3 reviews and has borderline scores, I requested a fourth review which is now available. Could authors respond to it, if they feel it necessary, and other reviewers also have a look at it?
Thanks a lot for the active and productive discussion so far.
Your area chair.

---

### Author Response · Authors · 2021-11-27
**Summary of All Manuscript Updates Made**

We authors have finished curating what we believe to be all major changes/additions requested by each reviewer. For convenient reference, below is a list of all such changes made to our manuscript since the beginning of this review process. Once again, we authors would like to thank all reviewers and area chairs for their feedback and support during this review process.

---

1. We have added a new set of experiments on Docking Benchmark 5 (DB5), a widely-used and challenging dataset for interface contact predictors.

2. All of our experiments, including those on DB5, now report both the top-k precision and the top-k metrics frequently found in combination with each other (e.g., when assessing the ranking performance of recommender systems).

3. We have rewritten Section 4 (Methods) in the manuscript to more clearly highlight the novelties of our new Geometric Transformer compared to the Graph Transformer or other Transformer-like architectures, using equations (https://openreview.net/forum?id=CS4463zx6Hi&noteId=plBIrAuD5k7) and revised figures to illustrate our contributions.

4. We have added two ablation studies (https://openreview.net/forum?id=CS4463zx6Hi&noteId=K9x9kFPMxXT and https://openreview.net/forum?id=CS4463zx6Hi&noteId=8FFykBSMbmd) experimentally confirming the superiority of the Geometric Transformer for protein structures compared to the original Graph Transformer.

5. We updated the figures in the main text to simplify and (ideally) make them easier to read and understand.

6. We have added (Section A.5) a discussion of the use of invariance and equivariance for different tasks (https://openreview.net/forum?id=CS4463zx6Hi&noteId=ut-Vof7pbG).

7. We have softened our previous claim on the general applicability of the Geometric Transformer for 3D graphs.

8. We have added a description of the machine learning algorithm(s) employed in each of our baseline methods.

9. We have expanded on our rationale for including certain baseline methods in our benchmark results (https://openreview.net/forum?id=CS4463zx6Hi&noteId=JoFo9esjH3).

10. We have reindexed our references to protein geometric features we use within the Geometric Transformer such that, in all locations in the manuscript, they start from 1 (https://openreview.net/forum?id=CS4463zx6Hi&noteId=plBIrAuD5k7).

---

### Decision · Program_Chairs · 2022-01-20

**Decision:**

Accept (Poster)

**Comment:**

This paper presents a novel neural network architecture to predict interacting residues among two interacting proteins, and evaluates its performance on benchmarks. While the reviews were initially mixed, there has been a productive discussion and significant improvements in the paper during the discussion, including in particular much needed clarifications about the proposed methods, and more experimental results with an ablation study to better assess the benefits of various design choices. While no reviewer is willing to champion this paper as a "strong accept", due to the relatively modest novelty compared to existing methods, there is a consensus towards "weak accept" given the final quality of the work presented and potential usefulness of the method for the problem tackled.